# CROSS-TABLE PRETRAINING TOWARDS A UNIVERSAL FUNCTION SPACE FOR TABULAR DATA

## ABSTRACT

Tabular data from different tables exhibit significant diversity due to varied definitions and types of features, as well as complex inter-feature and feature-target relationships. Cross-dataset pretraining, which learns reusable patterns from upstream data to support downstream tasks, have shown notable success in various fields. Yet, when applied to tabular data prediction, this paradigm faces challenges due to the limited reusable patterns among diverse tabular datasets (tables) and the general scarcity of tabular data available for fine-tuning. In this study, we fill this gap by introducing a cross-table pretrained Transformer, XTFORMER, for versatile downstream tabular prediction tasks. Our methodology insight is pretraining XTFORMER to establish a *meta-function* space that encompasses all potential feature-target mappings. In pre-training, a variety of potential mappings are extracted from pre-training tabular datasets and are embedded into the *meta-function* space, and suited mappings are extracted from the *meta-function* space for downstream tasks by a specified coordinate positioning approach. Experiments show that, in 190 downstream tabular prediction tasks, our cross-table pretrained XTFORMER wins both XGBoost and Catboost on 137 (72%) tasks, and surpasses representative deep learning models FT-Transformer and the tabular pre-training approach XTab on 144 (76%) and 162 (85%) tasks.

## 1 INTRODUCTION

Tabular data, prevalent in various application scenarios like economics and healthcare (Joseph et al., 2022; Johnson et al., 2016; Afonso et al., 2019; Wang & Sun, 2022), has sparked increased interest in the deep learning community (Gorishniy et al., 2021; Arik & Pfister, 2021). Nevertheless, tabular data is typically collected by users for specific purposes within particular contexts, which often results in small dataset scales. This makes it challenging to train a robust deep learning model for tabular predictions, and the issue of tabular prediction in data-limited scenarios remains unresolved. We found that in such cases, most representative deep tabular models, such as FT-Transformer (Gorishniy et al., 2021), AutoInt (Song et al., 2019), and SAINT (Somepalli et al., 2021), are ineffective. Currently, data-efficient gradient-boosted decision trees (GBDTs), such as XGBoost (Chen & Guestrin, 2016), remain the go-to solutions when training data are limited. However, their performance is still suboptimal in extremely data-scarce scenarios (e.g., with only 50 or 20 labeled training samples). A specialized deep tabular model, TabPFN (Hollmann et al., 2022), has been proposed for data-limited scenarios, but it is restricted to classification tasks, excluding regression, which limits its applicability in real-world practice.

One special research avenue addressing this challenge is *cross-dataset pretraining and transfer*, wherein shared patterns are extracted from upstream data to support downstream dataset-specific supervised tasks[1]. This approach has shown remarkable success in image and text processing tasks (He et al., 2022; Radford et al., 2021; Raffel et al., 2020). However, unlike images from diverse datasets that share common feature patterns, target semantics, and feature-target relations, tabular datasets exhibit diverse definitions of features and targets and a lack of a "common" feature relation (Yoon et al., 2020), leading to diverse and complex prediction functions (Grinsztajn et al., 2022). Such divergence impedes the discovery and transfer of reusable mapping patterns between datasets (Wang & Sun, 2022; Yoon et al., 2020).

---

[1]In this paper, each tabular dataset pre-defines a feature-target prediction task.

While some studies (Somepalli et al., 2021; Yoon et al., 2020) introduced pretext tasks (*e.g.*, in self-supervision) for intra-table pretraining and demonstrated effectiveness, these approaches were constrained by the information within a single dataset, neglecting the opportunity to leverage valuable knowledge from other datasets. These approaches struggle to achieve significant performance improvements since tabular data within a single table is often insufficient. Thus, a few studies further explored the pretraining of deep tabular models on tabular datasets where the features overlap with those of the target datasets (i.e., downstream) or are within the same domain (Wang & Sun, 2022; Levin et al., 2022; Zhou et al., 2023). However, these approaches may limit the broader applicability of pretrained models, as they do not allow for transfer to scenarios outside the domain. Recently, XTab (Zhu et al., 2023) pioneered pretraining a deep tabular model for any downstream tabular datasets, but it does not fully address the diversity issues among datasets and executes direct transfer, only achieving limited performance gains.

**Our key insight in addressing this challenge is to create a function space where diverse functions for different upstream datasets can be systematically embedded during pretraining.** For a given target dataset, one can extract a suitable feature-target mapping function from this space, thus leveraging the embedded knowledge. In other words, since the diversity between feature spaces and label spaces among tabular prediction tasks makes it difficult to find one unified mapping function like those found in image model pretraining, we propose learning a function space with a rich set of mapping functions that can be selectively utilized for target tasks. This more flexible approach does not require learning a universal mapping function during the pretraining stage for all tasks; instead, it focuses on learning representative mapping functions that can be selectively used.

To achieve this, we introduce a novel neural layer called the *Calibratable Linear Layer* (CALINEAR). This layer creates a linear function space using a set of basis linear functions, allowing the formulation of different linear functions through varying coordinate positions, i.e., different weighted combinations of the basis linear functions. By combining CALINEAR with non-linear modules (e.g., self-attention), we build XTFORMER, which extends the linear function spaces into a higher dimensional and more universal function space. We term this space the *meta-function* space, which is proficient in formulating diverse and intricate non-linear functions. In essence, XTFORMER acts as a 'meta'-model. The basis functions in CALINEARs are unique and crucial components of XTFORMER. Once pre-trained, it only requires finding the combination coefficients of these basis functions, then XTFORMER can can collapse into a feature-target mapping model suited to downstream tasks. We term this special downstream task process as *task calibration*. Compared to training a deep tabular model from scratch, this approach requires determining only a small number of parameters using downstream data (just the combination coefficients), significantly reducing the need for large amounts of training data.

While we posit that the *meta-function* space has the capacity to encompass all potential dataset-suited feature-target mapping functions, it is crucial to acknowledge that it may not perfectly align with every dataset. Hence, although the performance is already excellent, after executing the *task calibration* step, we also perform a light fine-tuning step called *refinement* (typically 5 epochs). This step further optimizes the model to be more optimal, potentially exploring beyond the initially established *meta-function* space.

In summary, our contributions are:

1. For the first time, this paper introduces a method for unifying the expression of diverse tabular feature-target mapping functions by establishing a universal function space that systematically accommodates various tabular prediction functions. These functions can be easily retrieved by coordinate positioning in the space, thereby facilitating effective cross-table pretraining and downstream task transfer.

2. We introduce the novel CALINEAR, leveraging a set of linear functions to formulate a variety of linear functions. This is in contrast to the traditional linear layer that expresses only one linear function constrained by fixed parameters. With CALINEAR, the expressive capacity of XTFORMER is enhanced, enabling it to effectively model the *meta-function* space that encompasses a wide range of tabular prediction functions.

3. A **calibration & refinement** fine-tuning paradigm is introduced to adapt XTFORMER for downstream tasks, utilizing the established feature-target function space while attaining an improved model that extends beyond the boundaries of the *meta-function* space.

## 2 RELATED WORK

### 2.1 TABULAR PREDICTION MODELS

Tabular data is encountered ubiquitously across diverse domains (Joseph et al., 2022; Assefa et al., 2020; Borisov et al., 2022). Gradient boosting decision trees (GBDT) algorithms (*e.g.*, XGBoost (Chen & Guestrin, 2016)) are currently the most widely used tabular prediction models due to their robustness, data-efficiency, and high accuracy (Katzir et al., 2020; Grinsztajn et al., 2022). In view of the notable advancements observed in deep learning across diverse fields, particularly in computer vision and natural language processing (Vaswani et al., 2017; Srivastava et al., 2015; He et al., 2016), there has been a growing research interest in expanding this success to the tabular data prediction tasks. Some of these deep learning approaches have demonstrated competitive compared to traditional GBDT algorithms in terms of robustness and accuracy (Arik & Pfister, 2021; Katzir et al., 2020; Chen et al., 2022; Wang & Sun, 2022; Hollmann et al., 2022; Gorishniy et al., 2021; McElfresh et al., 2023; Yan et al., 2023). However, current deep tabular models still face several challenges: (i) deep tabular models, as rotational invariant approaches with a larger parameter space, necessitate substantial amounts of training data (Ng, 2004) in training. Consequently, these models may exhibit subpar results on small-scale tabular datasets (Grinsztajn et al., 2022). (ii) Since there is no "common" correlation structure in tabular data (Yoon et al., 2020), most current deep tabular models were still trained in supervised learning manner, and have not successfully leveraged the benefits of transferred knowledge from other tabular datasets.

### 2.2 PRETRAINING ON TABULAR DOMAIN

While the spatial and sequential correlations of features in images and texts remain consistent across different datasets (Yoon et al., 2020), tabular data exhibits no common correlations (Jain et al., 2021), presenting unique challenges for cross-table pre-training on tabular datasets. Though numerous pre-training methods have been proposed for tabular data (Arik & Pfister, 2021; Yoon et al., 2020; Somepalli et al., 2021; Wang & Sun, 2022; Zhu et al., 2023), the majority of these approaches focus on intra-table pre-training, which avoids confronting the heterogeneity between datasets but also forfeits the opportunity to leverage reusable knowledge from other datasets. Recently, some studies took a step forward, achieving transfer learning on tables with shared label and feature spaces (Wang & Sun, 2022; Fang et al., 2019; Li et al., 2021). Furthermore, Levin et al. (2022) proposed a pseudo-feature approach to align upstream and downstream feature sets for intra-domain table pre-training. However, these methods still struggle to handle arbitrary cross-table pre-training, and the constraints imposed by the relationships between upstream and downstream datasets hinder their broader applicability. Recently, XTab (Zhu et al., 2023) conducted the first attempt on cross-dataset pre-training. However, it did not fully resolve the this issue, consequently achieving outcomes only slightly better than the supervised baseline.

## 3 METHODOLOGY

Existing neural network models, constrained by fixed parameters post-training, can only accommodate a single and fixed mapping. Such methods are sufficient for pre-training and transfer on many modalities (e.g., images) (Radford et al., 2021), since their features and high-level semantics often share distributions across datasets. However, tabular datasets are heterogeneous, characterized by diverse feature and target spaces, as well as distinct inter-feature and feature-target relationships. Thus, it is challenging to utilize existing neural network structures, as reusable patterns across tabular datasets are not significant. Our methodology insight involves constructing a universal function space and systematically embedding all mappings into this space. This allows different mappings to be expressed as various combinations of the space's basis functions, and the basis function parameters are thus reused. To achieve this, we first introduce a new *Calibratable Linear layer* (CALINEAR) to create a linear function space that can articulate a broad spectrum of linear functions (Sec. 3.1). In Sec. 3.2, we depict how to build XTFORMER with CALINEAR to establish a *meta-function* space capable of expressing heterogeneous functions for different datasets.

### 3.1 CALINEAR: CALIBRATABLE LINEAR LAYER

Linear layers are the major building blocks of modern neural networks, including Transformer-like architectures. However, a classical linear layer with fixed parameters struggle to adapt to diverse linear functions, especially when the model is applied to a new tabular dataset. Retraining the model from scratch, while avoiding the mentioned issue, is susceptible to overfitting and insufficient training of parameters, as most tabular datasets are small. To address this dilemma, we propose a *calibratable linear layer* (CALINEAR) for easy expression of different linear functions.

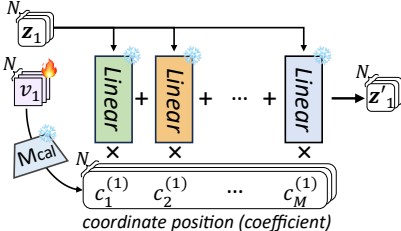

Figure 1: Illustrating Linear with a set of basis linear layers (Linear) and a *calibration module* ($\mathrm{M}_{cal}$). An element $v_n$ of $\mathbf{v} \in \mathbb{R}^N$ is input to $\mathrm{M}_{cal}$ to yield the coefficients $[c_1^{(n)}, ..., c_M^{(n)}]$ for the feature embedding $\mathbf{z}_n$. Only $\mathbf{v}$ is tuned for a new dataset.

Recall that any linear function $f$ in a linear function space $\mathcal{F}$ with a finite set of basis functions $\Phi = \{\phi_m\}_{m=1}^{M}$ can be expressed as a combination of these basis functions, by:

$$f(x; \mathbf{c}, \Phi) = \sum_{m=1}^{M} c_m \phi_m(x), \qquad (1)$$

where $c_m$ is a coefficient for $\phi_m$. Our main idea is to train the basic linear layers $\Phi$ during the pre-training phase, which will be shared across all the datasets. After this phase, for any new dataset, we can calibrate only the coefficient $\mathbf{c} = [c_1, \ldots, c_M] \in \mathbb{R}^M$ to obtain the final function $f$. Our objective is that during the pre-training phase, the basic linear layers will learn a meaningful function space, thus making it easier to find the final function $f$ parameterized by the coefficients $\mathbf{c}$. Assuming the input and the output dimensions of a linear layer are both $d$, training one basis linear layer amounts to tuning $d^2$ parameters using a fully-connected network (*e.g.,* $d = 256$ in FT-Transformer (Gorishniy et al., 2021); omitting the bias weights). In contrast, calibrating $c_m$ merely requires to determine $M$ parameters ($M = 4$ in our experiments).

Assuming the basis linear layers are fixed, a simple approach to determine coefficients $\mathbf{c}$ for fitting a target mapping is to separately train $M$ scalar parameters. To further reduce the number of parameter to calibrate, however, we introduce a simple *calibration module* ($\mathrm{M}_{cal}$) that generates these coefficients with common "anchor" context information specific to the target function. Specifically, for each CALINEAR layer, the associated $\mathrm{M}_{cal}$ is a simple two-layer multi-layer perceptron (MLP) followed by a softmax output. For any feature in a new tabular dataset, we allocate and optimize a randomly initialized context $v \in \mathbb{R}$, which is then passed through the $\mathrm{M}_{cal}$ to generate a set of coefficients $\mathbf{c} = [c_1, \ldots, c_M]$ to fit the target mapping:

$$\mathbf{c} = \mathrm{M}_{cal}(v) = \texttt{Softmax}(\texttt{MLP}(v)) \qquad (2)$$

Note that $\mathrm{M}_{cal}$ is jointly optimized with $v$ during the pre-training phase, while when fine-tuning for a target mapping, $\mathrm{M}_{cal}$ is frozen and only $v$ is optimized.

In short, a CALINEAR layer consists of $M$ basis learnable linear layers $\{\texttt{Linear}_m\}_{m=1}^{M}$ as well as a calibration module. Formally, given a learnable context $v \in \mathbb{R}$ (depending only on the dataset but shared across all CALINEAR layers in a model), with $\mathbf{z} \in \mathbb{R}^d$ denoting the input feature, the output $\mathbf{z}' \in \mathbb{R}^d$ of CALINEAR is given by:

$$\mathbf{z}' = \sum_{m=1}^{M} c_m \texttt{Linear}_m(\mathbf{z}), \qquad (3)$$

$$\text{where } \mathbf{c} = [c_1, \ldots, c_M] = \mathrm{M}_{cal}(v).$$

Note that the top of $\mathrm{M}_{cal}$ is a softmax layer ensuring that $\sum_{m=1}^{M} c_m = 1$ to control the magnitude of the CALINEAR output for stable learning dynamics. An illustration of CALINEAR can be found in Fig. 1. The $M$ basis linear layers and $\mathrm{M}_{cal}$ are *frozen* when tuning CALINEAR to fit a new function, and only $\mathbf{v}$ is tuned (see Sec. 3.3 for more details).

**Remarks:** While it may seem from Eq. (3) that a linear combination of the linear layers is still a linear layer, we should emphasize that the distinctive feature of the CALINEAR is the separate training of the basis linear layers $\Phi$ and the coefficients $\mathbf{c}$. The benefits are at least two-fold: **(i) capable of expressing a bunch of linear functions v.s. one single linear function after pretraining.** If the pretrained parameters in the linear layer(s) are kept fixed, a conventional linear layer can only

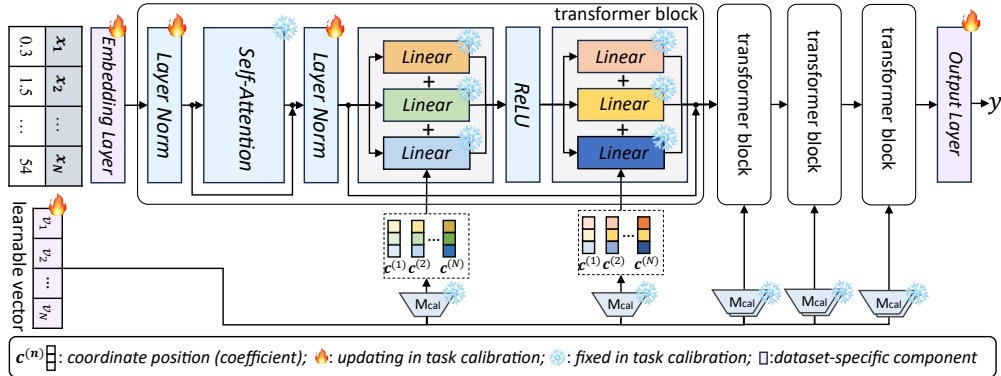

Figure 2: An illustration of XTFORMER framework. In the task calibration fine-tuning phase, the self-attention models and basis functions of the CALINEAR layers within the transformer blocks remain frozen; conversely, the dataset-specific components, learnable context vector **c** and the normalization layers undergo the task calibration fine-tuning step.

represent a single linear mapping, whereas CALINEAR can combine the basis linear functions $\Phi$ with different coefficients **c** to generate multiple linear mappings. **(ii) data-efficiency in fine-tuning.** If a fine-tuning is conducted, since only the context vector $v$ is used to generate $M$ coefficient to obtain a new mapping from the function space, CALINEAR thus allows the final XTFORMER (Sec. 3.2) to quickly calibrate to a new dataset that is potentially quite small. Obviously, determining just $M$ coefficients is much easier than fine-tuning all the parameters of a conventional linear layer.

## 3.2 XTFORMER ARCHITECTURE

As the CALINEAR layer only creates a linear function space, it is inadequate to express the diverse and non-linear dataset-suited functions, especially considering their typically irregular nature (Grinsztajn et al., 2022). Built upon CALINEARs, XTFORMER expands the linear function space into a more versatile *meta-function* space capable of modeling diverse and complex non-linear functions, by integrating various non-linear components (*e.g.*, activation functions and self-attention modules). The architecture of XTFORMER majorly follows a four-layer FT-Transformer (Gorishniy et al., 2021), but we replace the linear layers in the feedforward network (FFN) with CALINEAR Layer. Similar to XTab, we also use the dataset-specific embedding and output layer.

Formally, the FFN with CALINEAR layers is defined by:

$$\mathbf{z}' = \texttt{CaLinear}(\texttt{ReLU}(\texttt{CaLinear}(\mathbf{z}))) \tag{4}$$

$$= \sum_{m'=1}^{M} c_{m'} \texttt{Linear}_{m'}(\texttt{ReLU}(\sum_{m=1}^{M} c_m \texttt{Linear}_m(\mathbf{z}))) \tag{5}$$

That is, each FFN consists of two distinct CALINEARs, and the basis linear functions in these two CALINEARs are independent. Notably, for one dataset, all CALINEARs in the model share the same learnable context vector $\mathbf{v} = [v_1, \dots, v_N]$, as illustrated in Fig. 2. Note that though we only have one $\texttt{M}_{cal}$ in a CALINEAR, with different $v_n \in \mathbb{R}$, the coefficient **c** is still unique to each feature embedding.

In pretraining, the whole XTFORMER is trained on data from various datasets. When adapting it to a new dataset, only dataset-specific components (purple blocks in Fig. 2: embedding layer, output layer, and the learnable context vector **v**) as well as the normalization layers are fine-tuned. This allows XTFORMER to adapt to the diverse feature and target spaces of various tabular datasets.

The relation of CALINEAR and XTFORMER is twofold: (i) CALINEAR is a linear version of XTFORMER. While the CALINEAR can only create a linear function space, XTFORMER is able to establish a non-linear *meta-function* space that is more adept at capturing real-world feature-target relations. (ii) Stacking CALINEARs sequentially integrated with non-linear components, XTFORMER supports a broader function space. A single CALINEAR can be used to establish an $M$-dimensional space. For a XTFORMER with $L$ transformer blocks, there are $2ML$ sequentially stacked CALINEAR layers, enabling the model to encompass a function space of $M^{2L}$. In this paper, we set $M = 4$ and $L = 4$, resulting in the *meta-function* space with a theoretical dimensionality of up to 65,536.

### 3.3 PRETRAINING AND FINE-TUNING PARADIGM

We introduce a three-phase process for cross-table pretraining and transfer: (1) the cross-table pretraining and (2) a two-phase fine-tuning comprising (i) task calibration and (ii) refinement. In pretraining, XTFORMER learns on diverse tabular datasets to establish the *meta-function* space. Then, the task calibration phase aims to pinpoint the function best suited for the specific dataset within this *meta-function* space. Finally, in the refinement phase, we conduct further fine-tuning for all parameters to seek an improved function.

**Cross-Table Pretraining**    In the pretraining phase, we pretrain our XTFORMER on a large set of datasets $\mathcal{D}$. We train the XTFORMER with task-specific prediction objectives: we use the cross-entropy loss function for classification task and the mean squared error (MSE) loss function for regression. In each epoch of pretraining, we randomly choose a dataset $D_i \in \mathcal{D}$ and train the dataset-specific components (purple blocks in Fig. 2, including embedding layer, output layer, and the context $\mathbf{v}$) as well as the shared body of XTFORMER. As a result, all datasets collectively contribute to shaping the shared part of the XTFORMER, modeling the *meta-function* space applicable to any dataset and seamlessly making the shared components cooperative with any the dataset-specific component. This pretraining phase serves two primary purposes: (i) pretraining all the components of XTFORMER, especially each basis function of CALINEARs to establish the high-dimensional *meta-function* space; (ii) training the $\mathtt{M}_{cal}$ to effectively collaborate with XTFORMER, so as to proficiently identify an optimal functions for each given dataset from the *meta-function* space.

**Task Calibration**    The objective of the task calibration phase is to determine an optimal function within the *meta-function* space. After pretraining on diverse datasets, we assume that the *meta-function* space is well-established. In the task calibration phase, for a dataset (task) unseen in the pretraining phase, we freeze the shared part of XTFORMER and train the dataset-specific components from scratch to align the XTFORMER with the task prediction function, as depicted in Fig. 2. The normalization layers undergo updates in this phase as a standard configuration, following prior works (Houlsby et al., 2019). The training objective in the task calibration phase aligns with that of the pre-training phase: Either cross-entropy or MSE loss depending on the task. After the task calibration, we assemble shared and dataset-specific components to yield a proficient model for the given downstream task.

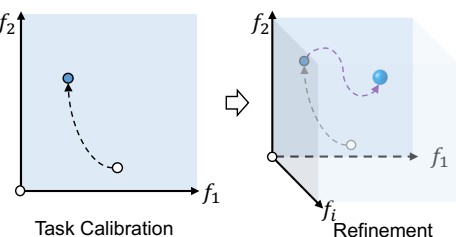

Figure 3: An illustration of task calibration and refinement. The task calibration (left) operates within the established function space to search for a dataset-suited model, while the refinement (right) optimizes all parameters for an improved model and may extend beyond the established space.

**Refinement**    Our assumption of the *meta-function* may still limit XTFORMER's ability to discover the optimal weights for a given dataset. The purpose of the refinement step is to further explore the "optimal" function that may exist beyond the *meta-function* space, as illustrated in Fig. 3. Continuing upon the the previous calibration step, we update all parameters (including the previously frozen layers) for a few epochs. Given the relatively limited sample size in a tabular dataset, we restrict the refinement epoch to a small number (5 by default) to mitigate over-fitting.

## 4    EXPERIMENTS

### 4.1    EXPERIMENTAL SETUP

We conduct experiments on a collection of tabular datasets derived from the OpenTabs V1 benchmark (Ye et al., 2023). Our data preprocessing and model training settings largely follow those of FT-Transformer (Gorishniy et al., 2021). The shared body of our model is a 4-layer Transformer encoder with an embedding size of 192 and 8 attention heads. We use the AdamW optimizer with a learning rate of $1 \times 10^{-4}$. All reported results are the average of 5 runs using different random seeds for statistical robustness.

Table 1: Performance comparison with presentative deep learning and GBDTs. Performance ranking (↓) within different settings are separately computed. The best results are marked in **bold** while the second and the third best results are underlined. "(t)", "(d)", and "(p)" respectively indicate "with hyperparameter tuned", "with default parameters", and "pretrained", respectively.

| Setting | XTFORMER (p) | XGboost(d) | XGboost(t) | Catboost(d) | Catboost(t) | FTT(d) | FTT(t) | AutoInt(d) | AutoInt(t) | SAINT(d) | XTab (p) |
|---|---|---|---|---|---|---|---|---|---|---|---|
| T-full | **3.84** ± 3.18 | 5.39 ± 2.92 | 4.00 ± 1.85 | 6.26 ± 2.83 | 4.26 ± 3.29 | 5.97 ± 2.42 | 3.97 ± 2.39 | 5.82 ± 3.31 | 4.68 ± 2.00 | 6.24 ± 3.06 | 7.58 ± 2.53 |
| T-200 | **2.84** ± 3.17 | 6.53 ± 3.60 | 4.76 ± 2.53 | 5.76 ± 2.93 | 5.21 ± 3.19 | 6.16 ± 2.84 | 6.05 ± 2.65 | 5.84 ± 2.22 | 5.95 ± 2.64 | 5.92 ± 2.71 | 6.84 ± 2.68 |
| T-100 | **2.50** ± 2.85 | 6.89 ± 3.37 | 4.42 ± 2.43 | 5.74 ± 2.80 | 5.11 ± 3.03 | 5.71 ± 2.51 | 5.39 ± 2.47 | 5.55 ± 2.54 | 5.61 ± 2.69 | 6.34 ± 2.91 | 6.74 ± 2.41 |
| T-50 | **3.16** ± 3.00 | 6.58 ± 3.45 | 5.71 ± 3.07 | 6.21 ± 2.70 | 5.50 ± 3.07 | 5.95 ± 3.09 | 4.76 ± 3.21 | 4.87 ± 2.89 | 5.13 ± 2.91 | 5.95 ± 2.86 | 6.92 ± 2.30 |
| T-20 | **2.08** ± 2.49 | 7.08 ± 3.22 | 6.29 ± 3.01 | 6.29 ± 3.27 | 5.53 ± 2.88 | 5.45 ± 3.01 | 5.34 ± 2.90 | 4.95 ± 2.65 | 6.05 ± 2.79 | 6.42 ± 2.83 | 6.32 ± 2.47 |

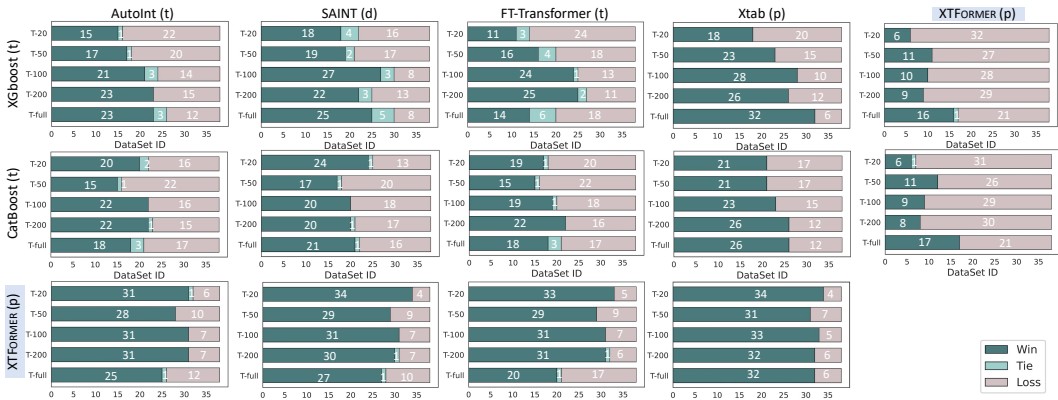

Figure 4: One-on-one comparison to assess the performance of XGBoost, Catboost and XTFORMER against representative deep learning approaches. XGBoost and CatBoost outperform or match existing deep learning approaches on a majority of datasets. However, our XTFORMER beats both XGboost and Catboost, revolutionizing the landscape of deep learning and GBDTs competition.

We compare our model against a comprehensive set of baselines, including dominant GBDTs (XGBoost, CatBoost), cutting-edge deep tabular models (FT-Transformer, SAINT), and a cross-table pretraining model (XTab). Following established practices (Zhu et al., 2023; Hollmann et al., 2022), we evaluate all methods by their performance ranking across datasets (Accuracy for classification and MSE for regression) and conduct a direct win-tie-loss analysis. Full implementation details, dataset specifics, and hyperparameter settings are provided in Section B.

## 4.2 PERFORMANCES AND DISCUSSION

**General performance** We compare XTFORMER against existing algorithms on 190 tasks (on 38 datasets across five different settings). The performance rankings are presented in Table 1 with the raw results in Section E. It is obvious that XTFORMER consistently outperforms all other methods in every setting, especially when the training and validation data are highly limited. Except for our approach, generally, deep learning approaches cannot beat hyperparameter-tuned XGboost and Catboost in most settings, corroborating the conclusions drawn in previous researches (Gorishniy et al., 2021; Grinsztajn et al., 2022). In order to further inspect whether our approach has altered the landscape of the tabular prediction domain, we conducted an one-on-one comparison for XTFORMER, XGBoost, and Catboost against previous deep learning models, as shown in Fig. 4. It is clear that XGBoost and Catboost outperform or achieve comparable performance to existing deep learning approaches across a majority of datasets and settings, while these deep learning approaches demonstrate superiority in fewer setting. In contrast, neither XGBoost nor Catboost surpasses XTFORMER, suggesting a transformative impact on the landscape of deep learning and GBDTs competition. One can observe that our model outperforms baseline approaches in the vast majority of tasks (typically over 70%). For example, it outperforms classic GBDTs, XGBoost and Catboost, on 137 (72%) tasks, FT-Transformer on 144 (76%) tasks, and the previous tabular pre-training model XTab on 162 (85%) tasks.

**Comparison with TabPFN** Across 20 classification datasets, under the 5 settings of T-full, T-200, T-100, T-50, and T-20, our XTFORMER outperforms TabPFN on 16, 14, 16, 14, and 12 tasks,

respectively—surpassing it in the majority of cases. No ties occurred. Therefore, overall, our approach consistently outperforms TabPFN in both full and small dataset scenarios.

**How many basis functions should we use?** The number of basis functions in CALIN-EAR directly impacts the dimensionality of the established *meta-function* space. Intuitively, a higher count of basis functions leads to a larger function space. In this study, we investigate the impact of varying the number of basis functions on model performance. We conduct experiments using 1, 2, 4, and 6 basis functions in each CALINEAR, re-pretraining the XTFORMER from scratch for each setting. Evaluation is performed on both the T-full and a data-insufficient setting (T-100), and performance rankings are computed separately. As shown in Table 2, the model exhibits improved performance with an increasing number of basis functions. Models with 4 or 6 basis functions markedly outperform those with 1 and 2. However, the difference between 4 and 6 basis functions is not significant. Actually, using 4 basis functions in XTFORMER has already yielded a substantial *meta-function* space with $n^{2L} = 4^8 = 65,536$ dimensions, which is sufficient to represent the majority of potential functions.

Table 2: Performance ranking ($\downarrow$) under T-full and T-100 settings, respectively, with different quantities of basis functions (BFs). The top performances are marked in **bold**, and the runner-ups are underlined.

| #. BF(s) | 1 | 2 | 4 | 6 |
|---|---|---|---|---|
| T-full | 2.47±1.14 | 2.43±1.01 | **2.10**±1.12 | 2.13±0.94 |
| T-100 | 2.87±1.01 | 2.43±1.01 | 2.20±1.19 | **2.17**±1.15 |

**How does the $\mathtt{M}_{cal}$ impact the results?** In this study, we devise a $\mathtt{M}_{cal}$ to process a learnable context vector $\mathbf{v}$ to generate coefficients, aiming to identify dataset-suited functions in the *meta-function* space. An intuitive alternative is to directly assign learnable coefficients for each CALINEAR and optimizing them to obtain coefficients. To compare these two approaches, we substitute the $\mathtt{M}_{cal}$ in the framework with learnable coefficients (also employing *softmax* like $\mathtt{M}_{cal}$) and present the comparative results in Table 3.

Remarkably, $\mathtt{M}_{cal}$ consistently outperforms the approach using vanilla learnable coefficients across all settings, outperforming on approximately 2/3 of the datasets. We attribute the observed superiority of using $\mathtt{M}_{cal}$ to two key factors: **(i) Coefficient Correlation Modeling**: $\mathtt{M}_{cal}$ can model the correlations among these coefficients, which is more effective for target model searching in the function space; **(ii) Parameter Efficiency**: $\mathtt{M}_{cal}$ allows us to optimize only $N$ learnable parameters ($\mathbf{v} \in \mathbb{R}^N$, $N$ is the number of feature). In contrast, the alternative method requires to learn $kN$ coefficients directly, where $k$ represents the number of CALINEAR ($k = 8$ in this paper). Given the potentially limited training data in tabular datasets, a reduced parameter count generally results in better performance.

Table 3: Ablation study results of $\mathtt{M}_{cal}$ vs. vanilla learnable coefficients. Win/tie/loss on 38 datasets in 5 settings are reported. The best performances are **bold**.

| Settings | T-20 | T-50 | T-100 | T-200 | T-full |
|---|---|---|---|---|---|
| #. $\mathtt{M}_{cal}$ win | **24** | **24** | **21** | **24** | **24** |
| #. tie | 2 | 1 | 2 | 1 | 2 |
| #. $\mathtt{M}_{cal}$ loss | 12 | 13 | 15 | 13 | 12 |

Table 4: Performance ranking ($\downarrow$) with varied epochs of task calibration (#.T) and refinement (#.R), under the **T-full** setting.

| #.T \ #.R | 0 | 3 | 5 | 10 |
|---|---|---|---|---|
| 40 | 9.71 ± 6.36 | 9.82 ± 5.76 | 9.61 ± 5.34 | 8.68 ± 5.68 |
| 80 | 7.74 ± 5.12 | 7.24 ± 4.80 | 7.76 ± 4.27 | 6.76 ± 4.40 |
| 120 | 6.82 ± 4.01 | 6.11 ± 4.04 | 6.32 ± 3.85 | 5.84 ± 4.00 |
| 160 | 6.03 ± 3.88 | 5.66 ± 3.74 | 4.58 ± 3.18 | 5.32 ± 3.58 |
| 240 | 5.61 ± 3.87 | 5.18 ± 3.72 | 4.37 ± 2.91 | 5.47 ± 3.86 |
| 320 | 5.58 ± 4.28 | 5.84 ± 5.17 | 5.37 ± 4.36 | 5.08 ± 4.83 |
| Ave | 6.91 ± 4.59 | 6.64 ± 4.54 | 6.33 ± 3.99 | 6.19 ± 4.39 |

Table 5: Performance ranking ($\downarrow$) with varied epochs of task calibration (#.T) and refinement (#.R), under the **T-100** setting.

| #.T \ #.R | 0 | 3 | 5 | 10 |
|---|---|---|---|---|
| 40 | 4.55 ± 3.80 | 4.82 ± 3.70 | 4.00 ± 3.55 | 4.32 ± 2.98 |
| 80 | 4.71 ± 2.86 | 4.84 ± 2.92 | 4.68 ± 2.74 | 4.68 ± 2.63 |
| 120 | 5.45 ± 2.84 | 5.61 ± 3.39 | 5.58 ± 3.35 | 5.29 ± 3.09 |
| 160 | 5.55 ± 3.26 | 5.53 ± 3.86 | 5.76 ± 3.46 | 5.92 ± 3.44 |
| Ave | 5.07 ± 3.19 | 5.20 ± 3.47 | 5.01 ± 3.27 | 5.05 ± 3.03 |

Table 6: Performance ranking ($\downarrow$) with varied epochs of task calibration (#.T) and refinement (#.R), under the **T-20** setting.

| #.T \ #.R | 0 | 3 | 5 | 10 |
|---|---|---|---|---|
| 40 | 4.55 ± 3.80 | 4.39 ± 3.51 | 4.42 ± 3.17 | 4.34 ± 2.90 |
| 80 | 4.84 ± 2.93 | 4.66 ± 2.75 | 4.82 ± 2.95 | 5.13 ± 2.96 |
| 120 | 5.39 ± 3.43 | 5.32 ± 3.20 | 4.76 ± 2.94 | 5.39 ± 3.54 |
| 160 | 5.26 ± 3.17 | 5.37 ± 3.28 | 5.61 ± 3.34 | 5.76 ± 3.39 |
| Ave | 5.01 ± 3.33 | 4.93 ± 3.18 | 4.90 ± 3.10 | 5.16 ± 3.20 |

**How does epoch number for pretraining, task calibration, and refinement impact results?** **(i) pretraining epoch.** We compare the results of the pretrained XTFORMER at different pretraining epochs with a baseline XTFORMER version without pretraining. Referring to Fig. 5, it is evident that pretraining yields a noteworthy performance enhancement over the first 50 pretraining epochs (per dataset). Following this, a gradual ascent phase occurs with the pretraining epoch number increases. Importantly, we note no significant performance decline, indicating the absence of both overfitting and collapse during pretraining, even as we constructed a *meta-function* space of up to 65,536 dimensions. **(ii) task calibration epoch.** We evaluate the impacts of fine-tuning step settings in the task calibration

phases and refinement phases, on original (T-full) and data-insufficient (T-20, T-100) scenarios. See Table 4, it is evident that on full datasets, as the number of task calibration steps increases, the model demonstrates an increasingly positive trend in performance. However, in the scenario of extreme data inefficiency, the trend is entirely opposite: referring to Table 5 and Table 6, an increase in the number of task calibration steps correlates with a decline in performance. This might be because the overly extensive fine-tuning process tends to result in overfitting when the training and validation samples are extremely limited. **(iii) refinement epoch.** Upon reviewing Table 4, 5, and 6, it is apparent that the refinement phase offers additional performance gains (compared to #.R=0), especially in the T-full setting. Besides, it is also apparent that our few-epoch refinement setting (5 epochs by default) does not hurt the performances even in highly data-insufficient scenarios (T-20 and T-100), implying the robustness of our approach.

It is apparent that the pre-trained *meta-function* space exhibits a degree of smoothness - features whose coefficients are close in the t-SNE space also exhibit similar feature-target distributions per (A-D). More importantly, upstream and downstream points close in the t-SNE plot also shows similar feature-target distributions. This provides insights into why XTFORMER works: For any new feature from a new dataset (a downstream red dot), $M_{cal}$ identifies good coordinates ($\mathbf{c}$) for it in the meta-function space such that its neighbors (nearby upstream black dots) are similar in terms of feature-target relation, as visibly confirmed in Fig. 6(A-D). Since $\mathbf{c}$ pinpoints the final feature-target function per Eq. (3), the resulting calibrated XTFORMER is also suitable for processing features similar to this new feature. In short, our approach performs an automatic clustering for feature-target functions, aiding in the discovery and utilization of similar mapping patterns in cross-table transfer. Another example is given in Section F.

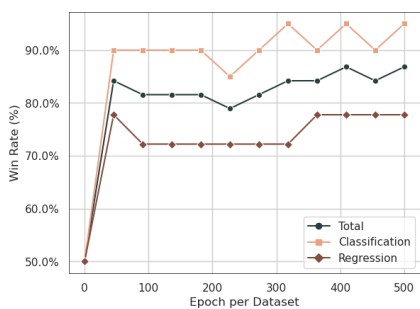

Figure 5: The win rate of the pre-trained XTFORMER in comparison to the baseline, across varying numbers of pretraining epochs.

**Training Time Comparison.** Due to the varying training strategies and extensive hyperparameter tuning required by parts of methods, we compared training times to evaluate how long each model takes to train on a single tabular prediction task. We cal-

Table 7: Model Development Time Comparison. The ranking ($\downarrow$) of average development time along with the standard deviation across all datasets is reported.

| XTFORMER | XGb(t) | Cat(t) | FTT(t) | AutoInt(t) | SAINT(d) | XTab |
|----------|--------|--------|--------|------------|----------|------|
| 3.45±1.87 | 3.00±1.21 | 4.84±1.83 | 6.34±1.24 | 4.89±0.82 | 3.26±1.52 | 1.80±1.12 |

culated the mean and standard deviation of training times' ranking across all datasets, as shown in Table 7. Although XTFORMER significantly outperforms other methods (see Figure 4 and Table 1), it also matches XGBoost in speed and outperforms deep learning models like FT-Transformer, AutoInt, and SAINT, as well as CatBoost, which all involve extensive hyperparameter tuning to obtain competitive performances. XTab is the fastest due to its lack of hyperparameter tuning and relatively few fine-tuning iterations. However, while XTab is the fastest, its performance may not meet the practical needs of all applications, which somewhat diminishes the value of its speed advantage. Note that these speed comparisons were conducted under the T-full setting, and differences between methods may be smaller in data-limited scenarios.

## 5 CONCLUSIONS

This paper presented XTFORMER to address the inherent challenge of heterogeneity in cross-table pre-training and transferability. The cross-table pretraining of XTFORMER serves to establish a universal function space, known as the *meta-function* space, enabling the systematic representation of diverse, dataset-specific, and intricate functions through coordinated positions within the space. When applied to downstream datasets, we can effectively find a task-suited functions by merely combining well-pretrained basis functions spanning the *meta-function* space. Experiments demonstrate that, our method outperforms all existing deep tabular prediction models by considerable margins, including dominant GBDTs, even with highly limited training data.

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

## A LLM USAGE

We made light use of a large language model (GPT-4, OpenAI) for grammar checking, spelling correction, and minor language polishing. The model was **not** involved in research ideation, experimental design, analysis, or interpretation. All scientific contributions and conclusions are entirely the responsibility of the authors.

## B EXPERIMENTAL SETUP

**Datasets**  We utilize labeled datasets from the high-quality large tabular data benchmark OpenTabs V1 (Ye et al., 2023). We systematically exclude datasets containing features with free text. Additionally, exclude duplicated datasets from the database, ensuring no overlap between the pretraining and downstream datasets.

**Implementation Details**  We employ the data preprocessing strategy following the FT-Transformer (Gorishniy et al., 2021) setting. The quantile transformation (Pedregosa et al., 2011) is applied onto raw features, and standardization is applied to regression targets. We consolidate all datasets into a unified dataloader, which allows to randomly sample a dataset per pretraining step.

Each dataset undergoes pretraining with the shared model body, accompanied by its dataset-specific embedding layer, output layer, and learnable context vector $\mathbf{v}$. We utilize a stack of 4 transformer blocks to construct the shared part of XTFORMER, with the feature embedding size of 192 and 8 attention heads. The number of basis linear layer in CALINEAR is set to 4 ($M = 4$). In both the pretraining and fine-tuning phases, we utilize the AdamW optimizer (Loshchilov & Hutter, 2018) with a learning rate set to 1e-4. Consistent with (Gorishniy et al., 2021), a weight decay of 1e-5 is applied to all components, excluding the embedding layer, layer normalization, and the bias terms. All experiments are executed on NVIDIA A100 GPUs 5 times with different random seeds, employing a batch size of 1024. When testing algorithms under limited dataset scenarios, the variation in random seeds also introduce changes to the sampled training and validation data. For pretraining, we execute an average of 350 epochs for each dataset. This involves incorporating a warm-up phase for the dataset-specific components during the initial 20% of steps, followed by a linear decay to zero. The default epoch amounts of task calibration for full dataset scenario and limited training data scenarios are 240 and 40, respectively. We set the default number of refinement epochs to 5. The best checkpoints on the validation set is used. For our XTFORMER, we only use hyperparameters without tuning.

**Baseline Algorithms**  We compare our XTFORMER with various representative supervised or pretrained tabular prediction models, including (i) the dominant GBDTs algorithms: XGBoost (Chen & Guestrin, 2016) and CatBoost (Prokhorenkova et al., 2018), (2) cutting-edge deep tabular models including FT-Transformer (FT-T) (Gorishniy et al., 2021), SAINT (Somepalli et al., 2021), AutoInt (Song et al., 2019), as well as (3) the open-source cross-table pretraining model XTab (Zhu et al., 2023). We also conducted a separate comparison with TabPFN (Hollmann et al., 2022), which, although excels in predictions for extremely small datasets, is limited to classification tasks only. All the algorithms trained through supervised learning undergo hyperparameter fine-tuning for better performances.

The hyper-parameter tuning settings follow their original configurations and the settings outlined in (Gorishniy et al., 2021).

**Metrics**  With the diversity of tabular datasets, no single model outperforms every competitor across all datasets (Gorishniy et al., 2021; Yan et al., 2023; Grinsztajn et al., 2022).

We thus compare model performances by: (i) performance ranking across all approaches, based on the accuracy for classification and the mean squared error for regression; following the practice in previous works (Zhu et al., 2023; Hollmann et al., 2022); and (ii) a win-tie-loss analysis for a direct comparison between two approaches.

## C  HYPERPARAMETER-TUNING SETTINGS

Due to the heterogeneity of tabular datasets, previous supervisedly trained deep learning and GBDTs algorithms require hyper-parameter tuning to achieve good performances, except for SAINT (Somepalli et al., 2021). For the compared AutoInt (Song et al., 2019), FT-Transformer (Gorishniy et al., 2021), XGboost (Chen & Guestrin, 2016), and Catboost (Prokhorenkova et al., 2018), we use Optuna library (Akiba et al., 2019) to conduct Bayesian optimization (the Tree-Structured Parzen Estimator algorithm) following (Gorishniy et al., 2021). All these approaches are tuned 100 iterations. For SAINT, we utilize the predefined configurations sourced from its original paper; for XTab (Zhu et al., 2023), we run the official code with the given configurations[2]. For TabPFN, we run the code following the given configurations in the paper. The hyper-parameter tuning spaces are reported below.

**XGboost.**  We implement XGboost by using the *XGBoost* python package[3]. Following (Gorishniy et al., 2021), we fix the following hyperparameters:

- `booster` = "gbtree";
- `early stopping rounds` = 50;
- `#. estimators` = 2000.

The rest hyperparameter are extensively tuned for each dataset, and the optimization spaces are listed below:

- `learning rate`: Log-Uniform distribution $[10^{-5}, 1]$;
- `max depth`: Discrete uniform distribution $[3, 10]$;
- `subsample`: Uniform distribution $[0.5, 1]$;
- `colsample by tree`: Uniform distribution $[0.5, 1]$;
- `colsample by level`: Uniform distribution $[0.5, 1]$;
- `min child weight`: Uniform distribution $[10^{-8}, 10^5]$;
- `alpha`: Uniform choice $\{0, \text{Log-Uniform distribution } [10^{-8}, 10^2]\}$;
- `lambda`: Uniform choice $\{0, \text{Log-Uniform distribution } [e^{-8}, 10^2]\}$;
- `gamma`: Uniform choice $\{0, \text{Log-Uniform distribution } [e^{-8}, 10^2]\}$.

**CatBoost.**  The catboost algorithm is implemented by using the *CatBoost* python package[4]. The following configurations are fixed:

- `early stopping rounds` = 50;
- `od-pval` = 0.001;
- `iterations` = 2000;
- `task type` = "GPU".

The hyperparameter tuning spaces are reported as below:

- `learning rate`: Log-Uniform distribution $[10^{-5}, 1]$;
- `max depth`: Discrete uniform distribution $[3, 10]$;
- `L2 leaf reg`: Log-Uniform distribution $[1, 10]$;
- `bagging temperature`: Uniform distribution $[0, 1]$;
- `leaf estimation iterations`: Discrete uniform distribution $[1, 10]$.

---

[2] `https://github.com/BingzhaoZhu/XTab`
[3] `https://xgboost.readthedocs.io/en/latest/index.html`
[4] `https://pypi.org/project/catboost/`

**FT-Transformer.** The hyperparameter optimization spaces are reported below:

- `the number of layers`: Discrete uniform distribution $[1, 4]$;
- `feature embedding size`: Discrete uniform distribution $[64, 512]$;
- `residual dropout`: Uniform distribution $[0, 0.2]$;
- `attention dropout`: Uniform distribution $[0, 0.5]$;
- `FFN dropout`: Uniform distribution $[0, 0.5]$;
- `FFN factor`: Uniform distribution $[\frac{2}{3}, \frac{8}{3}]$;
- `learning rate`: Log-Uniform distribution $[10^{-5}, 10^{-3}]$;
- `weight decay`: Log-Uniform distribution $[10^{-6}, 10^{-3}]$.

**AutoInt.** We utilize a modified version of AutoInt, as sourced from (Gorishniy et al., 2021), which has demonstrated superior performance compared to the original version.

- `the number of layers`: Discrete uniform distribution $[1, 6]$;
- `feature embedding size`: Discrete uniform distribution $[8, 64]$;
- `residual dropout`: Uniform distribution $[0, 0.2]$;
- `attention dropout`: Uniform distribution $[0, 0.5]$;
- `learning rate`: Log-Uniform distribution $[10^{-5}, 10^{-3}]$;
- `weight decay`: Log-Uniform distribution $[10^{-6}, 10^{-3}]$.

# D  DETAILED INFORMATION OF 38 DOWNSTREAM DATASETS

Here we summarize the key detailed information of 38 downstream datasets in Table 8. We use the same train-valid-test split for all the approaches.

Table 8: The details of the used 38 downstream tabular datasets. "#. Num" and "#. Cat" denote the numbers of numerical and categorical features, respectively. "#. Sample" presents the size of the dataset.

| Dataset | Abbr. | Task type | #. Num | #. Cat | #. Samples |
|---|---|---|---|---|---|
| train_2131_houses | HO | regression | 8 | 0 | 20640 |
| Customer_Churn_Records | CH | regression | 5 | 10 | 10000 |
| airline_passenger_satisfaction | PA | classification | 4 | 18 | 100000 |
| train_1487_tuiter | TU | regression | 1 | 39 | 761 |
| train_0431_visualizing_soil | VI | classification | 3 | 1 | 8641 |
| train_1648_Apple_Historical_Dataset | AP | regression | 5 | 0 | 9822 |
| train_0312_cpu_act | CP | classification | 21 | 0 | 8192 |
| train_0207_irish | IR | classification | 2 | 3 | 500 |
| train_0684_BNG_autoPrice_ | BN | regression | 15 | 0 | 100000 |
| train_0634_mozilla4 | MO | classification | 4 | 1 | 15545 |
| train_1879_The_2020_Pokemon_dataset | PO | classification | 10 | 26 | 1013 |
| train_0242_pm10 | PM | regression | 7 | 0 | 500 |
| train_1294_airlines | AL | classification | 3 | 4 | 26969 |
| accident_data | AC | classification | 6 | 25 | 100000 |
| train_1900_Another_Dataset_on_used_Fiat_500_1538_rows_ | FI | regression | 4 | 3 | 1538 |
| train_2644_concrete_compressive_strength | CO | regression | 7 | 1 | 1030 |
| bank_customer_survey | CU | classification | 7 | 9 | 45211 |
| train_1674_Violent_Crime_by_Counties_in_Maryland | CR | regression | 35 | 2 | 984 |
| train_1649_Tamilnadu_Crop_production | TA | regression | 1 | 5 | 13266 |
| Churn_Modelling | CM | classification | 5 | 6 | 10000 |
| train_1914_Hatred_on_Twitter_During_MeToo_Movement | HA | classification | 5 | 0 | 100000 |
| train_1618_depression | DE | classification | 11 | 10 | 1429 |
| train_1571_Temperature_Readings_IOT_Devices | TE | classification | 1 | 1 | 97606 |
| train_1770_Amazon_10Year_Stock_Data_Latest1997_2020 | AM | regression | 5 | 0 | 5852 |
| train_2706_abalone | AB | regression | 7 | 1 | 4177 |
| train_1564_Concrete | CC | regression | 7 | 1 | 1005 |
| train_1465_credit | CR | classification | 5 | 5 | 16714 |
| train_1414_AI4I2020 | AI | classification | 5 | 6 | 10000 |
| train_0247_boston | BO | regression | 11 | 2 | 506 |
| train_2743_Tallo | TL | regression | 11 | 9 | 100000 |
| train_2140_MiamiHousing2016 | MI | regression | 12 | 1 | 13932 |
| b_depressed | BD | classification | 11 | 10 | 1429 |
| train_1829_1000_Cameras_Dataset | CD | regression | 9 | 2 | 1038 |
| train_1251_AqSolDB_A_curated_aqueous_solubility_dataset | AQ | regression | 13 | 7 | 9982 |
| train_0611_flags | FL | classification | 2 | 26 | 194 |
| train_1564_Mammographic_Mass_Data_Set | MA | classification | 1 | 4 | 830 |
| train_0526_colleges_aaup | CA | classification | 13 | 2 | 1161 |
| train_0391_veteran | VE | classification | 3 | 4 | 137 |

# E  RAW RESULTS ON VARIOUS DATASETS UNDER DIFFERENT SETTINGS

In the main paper, we evaluate various approaches using a comprehensive metric performance ranking to offer an overall assessment of performance across various datasets. Here, we present detailed results for each model on diverse datasets, spanning 5 different settings, in Table 9, Table 10, Table 11, Table 12, and Table 13. The reported results represent averaged scores over 5 evaluations with different random seeds.

# F  COEFFICIENT-FUNCTION RELATIONSHIP VISUALIZATION

Recall that our calibration step aims to find a good transformation from embedding layer to the output layer (i.e. a good feature-target function). In Fig. 6(E), we collect coefficients for various features from both upstream (learned in pretraining) and downstream (before refinement) tables, and use t-SNE to visualize their distribution. Each point represents one feature (*i.e.* one feature-target relation) from one tabular dataset. In Fig. 6(A-D), we visualize the feature-target distribution of a

Table 9: Detailed performances of our XTFORMER and baseline algorithms, under the T-full setting. We prepend a "-" sign to the standardized MSE value (0-1) for the regression tasks, and thus all the results are the higher the better. "(t)", "(d)", and "(p)" respectively indicate "with hyperparameter tuned", "with default parameters", and "pretrained", respectively.

| Dataset | XTFORMER (p) | XGboost(d) | XGboost(t) | Catboost(d) | Catboost(t) | FTT(d) | FTT(t) | AutoInt(d) | AutoInt(t) | SAINT(d) | XTab (p) |
|---|---|---|---|---|---|---|---|---|---|---|---|
| FL | 0.809 | 0.807 | 0.749 | 0.742 | 0.742 | 0.710 | 0.903 | 0.802 | 0.774 | 0.903 | 0.825 |
| CO | -0.075 | -0.061 | -0.090 | -0.153 | -0.150 | -0.080 | -0.059 | -0.135 | -0.071 | -0.074 | -0.080 |
| CH | -0.264 | -0.258 | -0.256 | -0.255 | -0.254 | -0.255 | -0.254 | -0.266 | -0.255 | -0.255 | -0.275 |
| AB | -0.078 | -0.080 | -0.078 | -0.081 | -0.078 | -0.079 | -0.078 | -0.099 | -0.078 | -0.076 | -0.080 |
| HO | -0.070 | -0.065 | -0.066 | -0.068 | -0.062 | -0.072 | -0.066 | -0.110 | -0.071 | -0.070 | -0.071 |
| MO | 0.963 | 0.066 | 0.325 | 0.066 | 0.066 | 0.324 | 0.325 | 0.885 | 0.613 | 0.242 | 0.311 |
| CM | 0.857 | 0.148 | 0.796 | 0.203 | 0.256 | 0.204 | 0.204 | 0.790 | 0.399 | 0.208 | 0.190 |
| HA | 0.547 | 0.454 | 0.504 | 0.457 | 0.467 | 0.504 | 0.504 | 0.526 | 0.504 | 0.504 | 0.486 |
| AP | -0.033 | -0.034 | -0.034 | -0.036 | -0.034 | -0.037 | -0.037 | -0.037 | -0.037 | -0.037 | -0.039 |
| BN | -0.079 | -0.082 | -0.080 | -0.085 | -0.079 | -0.080 | -0.096 | -0.100 | -0.080 | -0.086 | -0.084 |
| MA | 0.884 | 0.850 | 0.880 | 0.677 | 0.684 | 0.865 | 0.880 | 0.681 | 0.872 | 0.857 | 0.814 |
| PM | -0.161 | -0.141 | -0.149 | -0.155 | -0.139 | -0.185 | -0.176 | -0.191 | -0.166 | -0.220 | -0.185 |
| TL | -0.003 | -0.002 | -0.003 | -0.003 | -0.002 | -0.006 | -0.004 | -0.004 | -0.004 | -0.006 | -0.006 |
| MI | -0.042 | -0.042 | -0.041 | -0.041 | -0.038 | -0.043 | -0.042 | -0.077 | -0.047 | -0.044 | -0.046 |
| CD | -0.080 | -0.075 | -0.064 | -0.059 | -0.047 | -0.077 | -0.062 | -0.068 | -0.076 | -0.080 | -0.080 |
| AL | 0.684 | 0.378 | 0.543 | 0.433 | 0.449 | 0.388 | 0.457 | 0.407 | 0.474 | 0.382 | 0.357 |
| CC | -0.062 | -0.062 | -0.086 | -0.145 | -0.144 | -0.084 | -0.065 | -0.082 | -0.065 | -0.096 | -0.085 |
| PA | 0.995 | 0.088 | 0.504 | 0.336 | 0.382 | 0.070 | 0.080 | 0.177 | 0.259 | 0.064 | 0.068 |
| CR | -0.102 | -0.097 | -0.097 | -0.097 | -0.093 | -0.104 | -0.096 | -0.092 | -0.101 | -0.104 | -0.106 |
| AQ | -0.004 | -0.005 | -0.005 | -0.007 | -0.005 | -0.006 | -0.006 | -0.005 | -0.005 | -0.006 | -0.007 |
| TE | 0.866 | 0.809 | 0.809 | 0.809 | 0.809 | 0.807 | 0.809 | 0.788 | 0.809 | 0.811 | 0.744 |
| AI | 0.904 | 0.999 | 0.999 | 0.999 | 0.999 | 0.999 | 0.999 | 0.999 | 0.999 | 0.999 | 0.963 |
| CP | 0.985 | 0.084 | 0.696 | 0.086 | 0.107 | 0.304 | 0.304 | 0.714 | 0.439 | 0.304 | 0.297 |
| BD | 0.550 | 0.830 | 0.835 | 0.838 | 0.843 | 0.834 | 0.836 | 0.642 | 0.834 | 0.834 | 0.760 |
| CU | 0.922 | 0.907 | 0.905 | 0.902 | 0.902 | 0.912 | 0.913 | 0.908 | 0.910 | 0.910 | 0.850 |
| BO | -0.075 | -0.074 | -0.072 | -0.054 | -0.047 | -0.092 | -0.070 | -0.095 | -0.087 | -0.090 | -0.093 |
| TA | -0.022 | -0.007 | -0.008 | -0.014 | -0.014 | -0.010 | -0.006 | -0.013 | -0.011 | -0.012 | -0.011 |
| IR | 1.000 | 0.025 | 0.500 | 0.350 | 0.413 | 0.500 | 0.500 | 0.710 | 0.638 | 0.500 | 0.486 |
| TU | -0.058 | -0.015 | -0.046 | -0.258 | -0.250 | -0.026 | -0.004 | -0.184 | -0.046 | -0.058 | -0.027 |
| FI | -0.091 | -0.087 | -0.087 | -0.090 | -0.085 | -0.089 | -0.086 | -0.086 | -0.088 | -0.090 | -0.094 |
| VE | 0.749 | 0.018 | 0.544 | 0.503 | 0.703 | 0.583 | 0.152 | 0.080 | 0.129 | 0.721 | 0.720 |
| CA | 1.000 | 0.043 | 0.274 | 0.043 | 0.043 | 0.726 | 0.726 | 0.914 | 0.780 | 0.274 | 0.685 |
| CR | 0.823 | 0.261 | 0.514 | 0.302 | 0.339 | 0.274 | 0.301 | 0.691 | 0.507 | 0.270 | 0.258 |
| DE | 0.585 | 0.830 | 0.833 | 0.834 | 0.843 | 0.834 | 0.834 | 0.754 | 0.834 | 0.834 | 0.814 |
| PO | 0.943 | 0.994 | 0.994 | 0.975 | 0.994 | 0.938 | 1.000 | 0.951 | 0.938 | 0.938 | 0.928 |
| AC | 1.000 | 0.000 | 0.504 | 0.001 | 0.005 | 0.000 | 0.013 | 0.106 | 0.345 | 0.000 | 0.000 |
| VI | 1.000 | 1.000 | 1.000 | 0.985 | 0.989 | 1.000 | 1.000 | 0.550 | 1.000 | 1.000 | 0.911 |
| AM | -0.061 | -0.057 | -0.058 | -0.060 | -0.057 | -0.065 | -0.065 | -0.055 | -0.057 | -0.065 | -0.070 |

cluster of coefficients, by plotting the mean activation after the embedding layer (denoted as $\overline{z}_{in}$) against that of the embedding before the dataset-specific output layer (denoted as $\overline{z}_{out}$)[5].

Similar to Fig. 6, here we provide an additional example to illustrate how coefficients represent a function in the *meta-function* space in Fig. 7, for reference.

---

[5]Note that we are actually interested in the relation of the feature and target *vectors* (input and output of the shared body of XTFORMER), but we plot only the mean of these vectors ($\overline{z}_{out}$ vs $\overline{z}_{in}$) because visualizing the relation of vectors is hard.

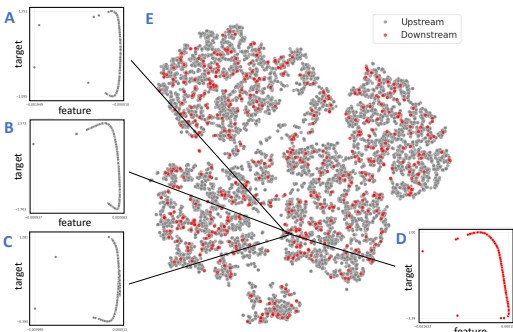

Figure 6: A-D: average activation of the embedding layer ($\overline{z}_{in}$ vs before the output layer ($\overline{z}_{out}$) - a proxy visualization of the feature-target relation we aim to learn or calibrate to. E: t-SNE visualization of the coefficients. Each dot in (A-D) represents a data (row) and in (E), a feature (column). Features whose coefficients in a cluster in (E) exhibit similar feature-target distribution in (A-D), implying that our calibration module is able to learn a suited feature-target function by finding the correct coefficients **c**.

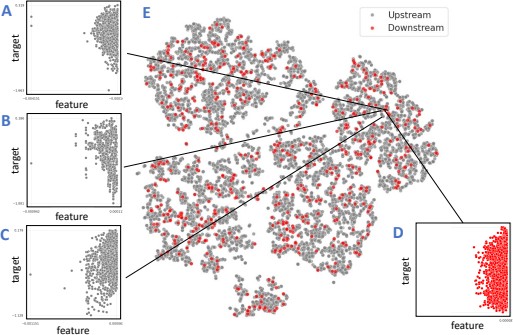

Figure 7: A-D: average activation of the embedding layer ($\overline{z}_{in}$ vs before the output layer ($\overline{z}_{out}$) - a proxy visualization of the feature-target relation we aim to learn or calibrate to. E: t-SNE visualization of the coefficients. Each dot in (A-D) represents a row and in (E), a feature. Features whose coefficients form a cluster in (E) also exhibit similar feature-target distribution in (A-D), indicating that our calibration module is able to learn a suitable feature-target function by finding the correct coefficients **c**.

Table 10: Detailed performances of our XTFORMER and baseline algorithms, under the T-200 setting. We prepend a "-" sign to the standardized MSE value (0-1) for the regression tasks, and thus all the results are the higher the better. "(t)", "(d)", and "(p)" respectively indicate "with hyperparameter tuned", "with default parameters", and "pretrained", respectively.

| Dataset | XTFORMER (p) | XGboost(d) | XGboost(t) | Catboost(d) | Catboost(t) | FTT(d) | FTT(t) | AutoInt(d) | AutoInt(t) | SAINT(d) | XTab (p) |
|---------|--------------|------------|------------|-------------|-------------|--------|--------|------------|------------|----------|----------|
| FL | 0.811 | 0.059 | 0.789 | 0.315 | 0.431 | 0.499 | 0.652 | 0.486 | 0.745 | 0.656 | 0.637 |
| CO | -0.081 | -0.085 | -0.075 | -0.155 | -0.151 | -0.122 | -0.126 | -0.079 | -0.135 | -0.119 | -0.121 |
| CH | -0.268 | -0.287 | -0.267 | -0.255 | -0.255 | -0.272 | -0.275 | -0.266 | -0.259 | -0.265 | -0.270 |
| AB | -0.083 | -0.093 | -0.088 | -0.091 | -0.088 | -0.099 | -0.092 | -0.099 | -0.085 | -0.100 | -0.103 |
| HO | -0.085 | -0.123 | -0.105 | -0.103 | -0.093 | -0.115 | -0.098 | -0.110 | -0.096 | -0.163 | -0.122 |
| MO | 0.954 | 0.932 | 0.921 | 0.904 | 0.938 | 0.672 | 0.907 | 0.885 | 0.899 | 0.880 | 0.847 |
| CM | 0.847 | 0.799 | 0.819 | 0.783 | 0.789 | 0.791 | 0.800 | 0.790 | 0.790 | 0.790 | 0.773 |
| HA | 0.544 | 0.357 | 0.396 | 0.442 | 0.504 | 0.439 | 0.524 | 0.492 | 0.411 | 0.376 | 0.411 |
| AP | -0.035 | -0.042 | -0.037 | -0.037 | -0.037 | -0.046 | -0.039 | -0.037 | -0.037 | -0.045 | -0.049 |
| BN | -0.087 | -0.114 | -0.099 | -0.101 | -0.101 | -0.101 | -0.105 | -0.100 | -0.101 | -0.103 | -0.102 |
| MA | 0.882 | 0.783 | 0.801 | 0.681 | 0.681 | 0.755 | 0.848 | 0.754 | 0.682 | 0.764 | 0.736 |
| PM | -0.158 | -0.114 | -0.105 | -0.118 | -0.100 | -0.187 | -0.151 | -0.191 | -0.132 | -0.191 | -0.204 |
| TL | -0.006 | -0.008 | -0.008 | -0.008 | -0.008 | -0.008 | -0.007 | -0.007 | -0.007 | -0.006 | -0.006 |
| MI | -0.054 | -0.083 | -0.067 | -0.075 | -0.068 | -0.083 | -0.079 | -0.077 | -0.077 | -0.081 | -0.082 |
| CD | -0.083 | -0.058 | -0.057 | -0.056 | -0.053 | -0.057 | -0.054 | -0.054 | -0.058 | -0.056 | -0.054 |
| AL | 0.676 | 0.411 | 0.393 | 0.443 | 0.443 | 0.394 | 0.419 | 0.407 | 0.410 | 0.379 | 0.369 |
| CC | -0.066 | -0.098 | -0.082 | -0.157 | -0.159 | -0.077 | -0.124 | -0.101 | -0.144 | -0.068 | -0.073 |
| PA | 0.988 | 0.618 | 0.757 | 0.961 | 0.622 | 0.898 | 0.653 | 0.654 | 0.817 | 0.656 | 0.868 |
| CR | -0.101 | -0.095 | -0.092 | -0.092 | -0.092 | -0.092 | -0.093 | -0.096 | -0.094 | -0.093 | -0.098 |
| AQ | -0.005 | -0.012 | -0.011 | -0.015 | -0.011 | -0.007 | -0.012 | -0.012 | -0.006 | -0.009 | -0.007 |
| TE | 0.856 | 0.800 | 0.767 | 0.800 | 0.800 | 0.788 | 0.806 | 0.788 | 0.788 | 0.788 | 0.752 |
| AI | 0.841 | 0.208 | 0.637 | 0.264 | 0.584 | 0.358 | 0.237 | 0.591 | 0.821 | 0.687 | 0.671 |
| CP | 0.970 | 0.856 | 0.714 | 0.824 | 0.832 | 0.714 | 0.316 | 0.714 | 0.282 | 0.714 | 0.682 |
| BD | 0.527 | 0.818 | 0.323 | 0.800 | 0.814 | 0.491 | 0.175 | 0.642 | 0.723 | 0.825 | 0.802 |
| CU | 0.903 | 0.547 | 0.643 | 0.643 | 0.849 | 0.861 | 0.722 | 0.635 | 0.621 | 0.725 | 0.813 |
| BO | -0.075 | -0.076 | -0.082 | -0.082 | -0.083 | -0.119 | -0.081 | -0.095 | -0.085 | -0.113 | -0.119 |
| TA | -0.022 | -0.024 | -0.024 | -0.024 | -0.024 | -0.024 | -0.022 | -0.023 | -0.023 | -0.023 | -0.025 |
| IR | 1.000 | 0.960 | 0.590 | 0.660 | 0.750 | 0.410 | 0.410 | 0.710 | 0.560 | 0.590 | 0.559 |
| TU | -0.056 | -0.012 | -0.019 | -0.250 | -0.249 | -0.042 | -0.180 | -0.067 | -0.212 | -0.090 | -0.045 |
| FI | -0.094 | -0.087 | -0.084 | -0.089 | -0.095 | -0.082 | -0.089 | -0.090 | -0.096 | -0.094 | -0.085 |
| VE | 0.719 | 0.050 | 0.625 | 0.701 | 0.680 | 0.615 | 0.446 | 0.132 | 0.624 | 0.051 | 0.559 |
| CA | 0.999 | 0.978 | 0.983 | 0.987 | 0.991 | 0.741 | 0.944 | 0.914 | 0.892 | 0.961 | 0.911 |
| CR | 0.813 | 0.250 | 0.264 | 0.272 | 0.293 | 0.329 | 0.779 | 0.426 | 0.744 | 0.391 | 0.359 |
| DE | 0.539 | 0.758 | 0.161 | 0.758 | 0.779 | 0.161 | 0.161 | 0.754 | 0.418 | 0.839 | 0.759 |
| PO | 0.914 | 0.951 | 0.960 | 0.951 | 0.951 | 0.951 | 0.941 | 0.951 | 0.951 | 0.951 | 0.913 |
| AC | 1.000 | 0.099 | 0.608 | 0.560 | 0.518 | 0.963 | 0.548 | 0.895 | 0.135 | 0.842 | 0.913 |
| VI | 1.000 | 0.993 | 0.550 | 0.858 | 0.862 | 0.450 | 0.550 | 0.550 | 0.453 | 0.550 | 0.514 |
| AM | -0.062 | -0.056 | -0.054 | -0.055 | -0.054 | -0.058 | -0.055 | -0.055 | -0.056 | -0.061 | -0.064 |

Table 11: Detailed performances of our XTFORMER and baseline algorithms, under the T-100 setting. We prepend a "-" sign to the standardized MSE value (0-1) for the regression tasks, and thus all the results are the higher the better. "(t)", "(d)", and "(p)" respectively indicate "with hyperparameter tuned", "with default parameters", and "pretrained", respectively.

| Dataset | XTFORMER (p) | XGboost(d) | XGboost(t) | Catboost(d) | Catboost(t) | FTT(d) | FTT(t) | AutoInt(d) | AutoInt(t) | SAINT(d) | XTab (p) |
|---|---|---|---|---|---|---|---|---|---|---|---|
| FL | 0.766 | 0.947 | 0.921 | 0.790 | 0.711 | 0.711 | 0.868 | 0.816 | 0.842 | 0.868 | 0.820 |
| CO | -0.074 | -0.094 | -0.102 | -0.158 | -0.153 | -0.133 | -0.135 | -0.131 | -0.154 | -0.146 | -0.142 |
| CH | -0.273 | -0.280 | -0.266 | -0.264 | -0.271 | -0.254 | -0.269 | -0.254 | -0.259 | -0.258 | -0.275 |
| AB | -0.079 | -0.085 | -0.082 | -0.085 | -0.084 | -0.086 | -0.088 | -0.086 | -0.081 | -0.089 | -0.094 |
| HO | -0.070 | -0.124 | -0.107 | -0.113 | -0.104 | -0.115 | -0.111 | -0.118 | -0.105 | -0.162 | -0.118 |
| MO | 0.966 | 0.105 | 0.325 | 0.095 | 0.108 | 0.379 | 0.675 | 0.325 | 0.692 | 0.325 | 0.361 |
| CM | 0.850 | 0.729 | 0.492 | 0.782 | 0.736 | 0.476 | 0.267 | 0.800 | 0.286 | 0.800 | 0.776 |
| HA | 0.554 | 0.377 | 0.485 | 0.420 | 0.405 | 0.514 | 0.528 | 0.424 | 0.466 | 0.463 | 0.471 |
| AP | -0.032 | -0.044 | -0.041 | -0.041 | -0.041 | -0.041 | -0.040 | -0.041 | -0.041 | -0.051 | -0.042 |
| BN | -0.079 | -0.112 | -0.106 | -0.110 | -0.112 | -0.116 | -0.108 | -0.106 | -0.105 | -0.145 | -0.121 |
| MA | 0.891 | 0.151 | 0.139 | 0.361 | 0.361 | 0.289 | 0.145 | 0.133 | 0.145 | 0.133 | 0.273 |
| PM | -0.171 | -0.172 | -0.154 | -0.153 | -0.151 | -0.173 | -0.181 | -0.181 | -0.181 | -0.177 | -0.187 |
| TL | -0.003 | -0.010 | -0.007 | -0.006 | -0.005 | -0.007 | -0.007 | -0.007 | -0.008 | -0.005 | -0.005 |
| MI | -0.042 | -0.092 | -0.079 | -0.084 | -0.077 | -0.090 | -0.084 | -0.090 | -0.093 | -0.095 | -0.099 |
| CD | -0.077 | -0.063 | -0.061 | -0.060 | -0.056 | -0.060 | -0.061 | -0.060 | -0.060 | -0.056 | -0.060 |
| AL | 0.682 | 0.426 | 0.558 | 0.432 | 0.455 | 0.478 | 0.558 | 0.536 | 0.496 | 0.545 | 0.544 |
| CC | -0.050 | -0.119 | -0.105 | -0.170 | -0.200 | -0.081 | -0.084 | -0.169 | -0.057 | -0.185 | -0.088 |
| PA | 0.995 | 0.625 | 0.886 | 0.911 | 0.882 | 0.835 | 0.674 | 0.959 | 0.829 | 0.697 | 0.812 |
| CR | -0.099 | -0.096 | -0.094 | -0.092 | -0.092 | -0.092 | -0.090 | -0.092 | -0.093 | -0.092 | -0.097 |
| AQ | -0.004 | -0.025 | -0.024 | -0.026 | -0.025 | -0.019 | -0.015 | -0.009 | -0.024 | -0.022 | -0.020 |
| TE | 0.866 | 0.801 | 0.803 | 0.803 | 0.803 | 0.793 | 0.793 | 0.793 | 0.793 | 0.793 | 0.747 |
| AI | 0.823 | 0.292 | 0.631 | 0.433 | 0.723 | 0.797 | 0.544 | 0.719 | 0.809 | 0.471 | 0.779 |
| CP | 0.972 | 0.866 | 0.699 | 0.846 | 0.802 | 0.699 | 0.301 | 0.699 | 0.258 | 0.699 | 0.692 |
| BD | 0.551 | 0.246 | 0.821 | 0.200 | 0.267 | 0.179 | 0.512 | 0.179 | 0.821 | 0.179 | 0.163 |
| CU | 0.919 | 0.603 | 0.835 | 0.816 | 0.632 | 0.899 | 0.748 | 0.630 | 0.709 | 0.690 | 0.813 |
| BO | -0.071 | -0.078 | -0.074 | -0.082 | -0.069 | -0.091 | -0.092 | -0.089 | -0.084 | -0.219 | -0.091 |
| TA | -0.025 | -0.010 | -0.010 | -0.010 | -0.010 | -0.010 | -0.010 | -0.010 | -0.010 | -0.010 | -0.011 |
| IR | 1.000 | 0.960 | 0.960 | 0.770 | 0.690 | 0.930 | 0.890 | 0.950 | 0.930 | 0.940 | 0.908 |
| TU | -0.058 | -0.063 | -0.073 | -0.289 | -0.270 | -0.248 | -0.117 | -0.201 | -0.286 | -0.198 | -0.203 |
| FI | -0.096 | -0.097 | -0.089 | -0.100 | -0.092 | -0.121 | -0.104 | -0.123 | -0.103 | -0.102 | -0.112 |
| VE | 0.708 | 0.628 | 0.705 | 0.641 | 0.665 | 0.656 | 0.678 | 0.668 | 0.636 | 0.679 | 0.673 |
| CA | 0.999 | 0.961 | 0.961 | 0.966 | 0.935 | 0.711 | 0.892 | 0.914 | 0.858 | 0.901 | 0.858 |
| CR | 0.818 | 0.284 | 0.489 | 0.302 | 0.312 | 0.816 | 0.746 | 0.569 | 0.645 | 0.477 | 0.752 |
| DE | 0.533 | 0.783 | 0.783 | 0.804 | 0.797 | 0.821 | 0.811 | 0.821 | 0.811 | 0.821 | 0.768 |
| PO | 0.924 | 0.960 | 0.936 | 0.936 | 0.965 | 0.936 | 0.941 | 0.936 | 0.896 | 0.936 | 0.849 |
| AC | 1.000 | 0.351 | 0.925 | 0.495 | 0.937 | 0.441 | 0.754 | 0.401 | 0.834 | 0.451 | 0.443 |
| VI | 1.000 | 0.021 | 0.005 | 0.094 | 0.102 | 0.054 | 0.017 | 0.099 | 0.063 | 0.034 | 0.050 |
| AM | -0.057 | -0.068 | -0.065 | -0.065 | -0.065 | -0.067 | -0.066 | -0.065 | -0.066 | -0.068 | -0.071 |

Table 12: Detailed performances of our XTFORMER and baseline algorithms, under the T-50 setting. We prepend a "-" sign to the standardized MSE value (0-1) for the regression tasks, and thus all the results are the higher the better. "(t)", "(d)", and "(p)" respectively indicate "with hyperparameter tuned", "with default parameters", and "pretrained", respectively.

| Dataset | XTFORMER (p) | XGboost(d) | XGboost(t) | Catboost(d) | Catboost(t) | FTT(d) | FTT(t) | AutoInt(d) | AutoInt(t) | SAINT(d) | XTab (p) |
|---|---|---|---|---|---|---|---|---|---|---|---|
| FL | 0.754 | 0.868 | 0.816 | 0.737 | 0.711 | 0.316 | 0.816 | 0.790 | 0.763 | 0.790 | 0.729 |
| CO | -0.101 | -0.186 | -0.196 | -0.175 | -0.169 | -0.180 | -0.122 | -0.153 | -0.143 | -0.155 | -0.155 |
| CH | -0.263 | -0.301 | -0.308 | -0.298 | -0.305 | -0.262 | -0.332 | -0.364 | -0.295 | -0.342 | -0.267 |
| AB | -0.086 | -0.112 | -0.096 | -0.097 | -0.093 | -0.123 | -0.119 | -0.107 | -0.114 | -0.110 | -0.114 |
| HO | -0.113 | -0.126 | -0.118 | -0.120 | -0.117 | -0.121 | -0.119 | -0.111 | -0.105 | -0.163 | -0.122 |
| MO | 0.908 | 0.869 | 0.789 | 0.839 | 0.838 | 0.603 | 0.677 | 0.811 | 0.790 | 0.678 | 0.617 |
| CM | 0.686 | 0.790 | 0.789 | 0.716 | 0.786 | 0.799 | 0.799 | 0.799 | 0.799 | 0.799 | 0.721 |
| HA | 0.526 | 0.161 | 0.376 | 0.427 | 0.441 | 0.328 | 0.259 | 0.311 | 0.381 | 0.313 | 0.313 |
| AP | -0.037 | -0.037 | -0.035 | -0.036 | -0.035 | -0.045 | -0.047 | -0.037 | -0.048 | -0.047 | -0.046 |
| BN | -0.114 | -0.114 | -0.116 | -0.116 | -0.114 | -0.136 | -0.099 | -0.103 | -0.103 | -0.135 | -0.148 |
| MA | 0.879 | 0.193 | 0.524 | 0.283 | 0.319 | 0.506 | 0.524 | 0.235 | 0.687 | 0.524 | 0.475 |
| PM | -0.169 | -0.172 | -0.181 | -0.158 | -0.155 | -0.173 | -0.187 | -0.167 | -0.176 | -0.176 | -0.177 |
| TL | -0.007 | -0.014 | -0.006 | -0.006 | -0.006 | -0.010 | -0.007 | -0.012 | -0.012 | -0.008 | -0.008 |
| MI | -0.084 | -0.102 | -0.092 | -0.105 | -0.099 | -0.157 | -0.089 | -0.098 | -0.095 | -0.158 | -0.158 |
| CD | -0.112 | -0.077 | -0.072 | -0.067 | -0.062 | -0.065 | -0.070 | -0.073 | -0.072 | -0.075 | -0.066 |
| AL | 0.604 | 0.605 | 0.587 | 0.524 | 0.481 | 0.574 | 0.593 | 0.555 | 0.573 | 0.582 | 0.577 |
| CC | -0.101 | -0.142 | -0.122 | -0.175 | -0.171 | -0.167 | -0.103 | -0.120 | -0.127 | -0.123 | -0.134 |
| PA | 0.945 | 0.077 | 0.378 | 0.696 | 0.391 | 0.407 | 0.824 | 0.363 | 0.758 | 0.405 | 0.378 |
| CR | -0.109 | -0.090 | -0.093 | -0.091 | -0.089 | -0.090 | -0.093 | -0.089 | -0.093 | -0.091 | -0.097 |
| AQ | -0.011 | -0.017 | -0.015 | -0.019 | -0.017 | -0.016 | -0.011 | -0.012 | -0.016 | -0.012 | -0.012 |
| TE | 0.774 | 0.806 | 0.767 | 0.796 | 0.796 | 0.792 | 0.710 | 0.792 | 0.710 | 0.792 | 0.774 |
| AI | 0.897 | 0.545 | 0.551 | 0.558 | 0.554 | 0.599 | 0.844 | 0.552 | 0.671 | 0.610 | 0.578 |
| CP | 0.934 | 0.847 | 0.840 | 0.752 | 0.869 | 0.706 | 0.838 | 0.897 | 0.891 | 0.706 | 0.681 |
| BD | 0.554 | 0.772 | 0.674 | 0.804 | 0.811 | 0.856 | 0.856 | 0.856 | 0.846 | 0.856 | 0.842 |
| CU | 0.747 | 0.350 | 0.494 | 0.672 | 0.530 | 0.671 | 0.454 | 0.561 | 0.434 | 0.657 | 0.655 |
| BO | -0.124 | -0.132 | -0.147 | -0.153 | -0.164 | -0.161 | -0.151 | -0.184 | -0.167 | -0.169 | -0.161 |
| TA | -0.024 | -0.014 | -0.014 | -0.014 | -0.014 | -0.014 | -0.014 | -0.014 | -0.014 | -0.014 | -0.015 |
| IR | 1.000 | 0.970 | 0.920 | 0.710 | 0.770 | 0.630 | 0.970 | 0.750 | 0.960 | 0.780 | 0.712 |
| TU | -0.121 | -0.159 | -0.093 | -0.261 | -0.261 | -0.263 | -0.130 | -0.169 | -0.246 | -0.248 | -0.268 |
| FI | -0.109 | -0.123 | -0.097 | -0.119 | -0.106 | -0.101 | -0.118 | -0.099 | -0.117 | -0.122 | -0.105 |
| VE | 0.661 | 0.565 | 0.597 | 0.657 | 0.629 | 0.644 | 0.573 | 0.646 | 0.646 | 0.643 | 0.605 |
| CA | 0.996 | 0.112 | 0.711 | 0.112 | 0.108 | 0.315 | 0.711 | 0.289 | 0.672 | 0.289 | 0.285 |
| CR | 0.733 | 0.664 | 0.625 | 0.654 | 0.649 | 0.689 | 0.685 | 0.707 | 0.720 | 0.654 | 0.661 |
| DE | 0.486 | 0.839 | 0.733 | 0.832 | 0.891 | 0.902 | 0.902 | 0.902 | 0.877 | 0.902 | 0.861 |
| PO | 0.863 | 0.449 | 0.845 | 0.582 | 0.716 | 0.840 | 0.838 | 0.838 | 0.758 | 0.846 | 0.814 |
| AC | 1.000 | 0.732 | 0.393 | 0.077 | 0.374 | 0.000 | 0.834 | 0.822 | 0.329 | 0.503 | 0.487 |
| VI | 0.968 | 0.008 | 0.008 | 0.149 | 0.107 | 0.223 | 0.185 | 0.293 | 0.026 | 0.230 | 0.218 |
| AM | -0.070 | -0.066 | -0.079 | -0.067 | -0.063 | -0.064 | -0.062 | -0.063 | -0.062 | -0.067 | -0.070 |

Table 13: Detailed performances of our XTFORMER and baseline algorithms, under the T-20 setting. We prepend a "-" sign to the standardized MSE value (0-1) for the regression tasks, and thus all the results are the higher the better. "(t)", "(d)", and "(p)" respectively indicate "with hyperparameter tuned", "with default parameters", and "pretrained", respectively.

| Dataset | XTFORMER (p) | XGboost(d) | XGboost(t) | Catboost(d) | Catboost(t) | FTT(d) | FTT(t) | AutoInt(d) | AutoInt(t) | SAINT(d) | XTab (p) |
|---|---|---|---|---|---|---|---|---|---|---|---|
| FL | 0.751 | 0.658 | 0.684 | 0.737 | 0.658 | 0.395 | 0.316 | 0.684 | 0.263 | 0.684 | 0.653 |
| CO | -0.115 | -0.213 | -0.180 | -0.188 | -0.183 | -0.173 | -0.123 | -0.170 | -0.194 | -0.161 | -0.172 |
| CH | -0.269 | -0.267 | -0.258 | -0.296 | -0.292 | -0.271 | -0.295 | -0.269 | -0.261 | -0.311 | -0.280 |
| AB | -0.096 | -0.102 | -0.096 | -0.100 | -0.092 | -0.113 | -0.108 | -0.115 | -0.105 | -0.110 | -0.119 |
| HO | -0.117 | -0.140 | -0.131 | -0.140 | -0.139 | -0.125 | -0.120 | -0.127 | -0.141 | -0.178 | -0.135 |
| MO | 0.898 | 0.097 | 0.097 | 0.090 | 0.090 | 0.402 | 0.178 | 0.325 | 0.191 | 0.325 | 0.373 |
| CM | 0.438 | 0.438 | 0.438 | 0.438 | 0.438 | 0.438 | 0.438 | 0.438 | 0.438 | 0.438 | 0.430 |
| HA | 0.520 | 0.463 | 0.465 | 0.477 | 0.511 | 0.488 | 0.520 | 0.507 | 0.479 | 0.517 | 0.488 |
| AP | -0.038 | -0.047 | -0.046 | -0.048 | -0.046 | -0.049 | -0.053 | -0.051 | -0.053 | -0.049 | -0.052 |
| BN | -0.120 | -0.124 | -0.126 | -0.126 | -0.122 | -0.136 | -0.109 | -0.109 | -0.115 | -0.136 | -0.140 |
| MA | 0.889 | 0.608 | 0.404 | 0.633 | 0.633 | 0.723 | 0.404 | 0.404 | 0.422 | 0.404 | 0.702 |
| PM | -0.172 | -0.189 | -0.251 | -0.190 | -0.193 | -0.194 | -0.191 | -0.193 | -0.195 | -0.190 | -0.194 |
| TL | -0.007 | -0.025 | -0.028 | -0.012 | -0.008 | -0.020 | -0.012 | -0.020 | -0.013 | -0.016 | -0.016 |
| MI | -0.100 | -0.123 | -0.127 | -0.130 | -0.120 | -0.101 | -0.129 | -0.124 | -0.105 | -0.120 | -0.106 |
| CD | -0.111 | -0.092 | -0.090 | -0.087 | -0.086 | -0.085 | -0.085 | -0.085 | -0.090 | -0.090 | -0.090 |
| AL | 0.579 | 0.462 | 0.466 | 0.465 | 0.448 | 0.575 | 0.532 | 0.531 | 0.460 | 0.453 | 0.529 |
| CC | -0.123 | -0.161 | -0.158 | -0.190 | -0.191 | -0.186 | -0.147 | -0.175 | -0.181 | -0.174 | -0.191 |
| PA | 0.926 | 0.110 | 0.402 | 0.295 | 0.401 | 0.755 | 0.793 | 0.820 | 0.698 | 0.144 | 0.683 |
| CR | -0.108 | -0.112 | -0.111 | -0.102 | -0.106 | -0.110 | -0.105 | -0.123 | -0.108 | -0.119 | -0.119 |
| AQ | -0.009 | -0.028 | -0.027 | -0.029 | -0.028 | -0.029 | -0.014 | -0.015 | -0.026 | -0.014 | -0.016 |
| TE | 0.773 | 0.804 | 0.779 | 0.804 | 0.804 | 0.794 | 0.709 | 0.794 | 0.702 | 0.794 | 0.782 |
| AI | 0.898 | 0.267 | 0.726 | 0.447 | 0.426 | 0.325 | 0.392 | 0.796 | 0.312 | 0.395 | 0.382 |
| CP | 0.935 | 0.637 | 0.793 | 0.804 | 0.684 | 0.870 | 0.895 | 0.906 | 0.930 | 0.669 | 0.816 |
| BD | 0.551 | 0.249 | 0.309 | 0.214 | 0.200 | 0.179 | 0.175 | 0.175 | 0.267 | 0.175 | 0.162 |
| CU | 0.780 | 0.547 | 0.566 | 0.748 | 0.746 | 0.591 | 0.573 | 0.698 | 0.699 | 0.613 | 0.603 |
| BO | -0.125 | -0.162 | -0.145 | -0.168 | -0.149 | -0.189 | -0.210 | -0.203 | -0.207 | -0.220 | -0.196 |
| TA | -0.024 | -0.013 | -0.013 | -0.013 | -0.013 | -0.013 | -0.013 | -0.013 | -0.013 | -0.013 | -0.014 |
| IR | 1.000 | 0.850 | 0.740 | 0.590 | 0.610 | 0.400 | 0.800 | 0.960 | 0.660 | 0.860 | 0.786 |
| TU | -0.133 | -0.208 | -0.272 | -0.284 | -0.283 | -0.142 | -0.214 | -0.253 | -0.205 | -0.269 | -0.146 |
| FI | -0.106 | -0.129 | -0.164 | -0.208 | -0.148 | -0.114 | -0.120 | -0.140 | -0.172 | -0.131 | -0.121 |
| VE | 0.632 | 0.121 | 0.158 | 0.534 | 0.337 | 0.510 | 0.411 | 0.393 | 0.615 | 0.531 | 0.505 |
| CA | 0.981 | 0.698 | 0.935 | 0.905 | 0.940 | 0.237 | 0.785 | 0.879 | 0.629 | 0.767 | 0.742 |
| CR | 0.749 | 0.651 | 0.552 | 0.489 | 0.536 | 0.746 | 0.567 | 0.658 | 0.657 | 0.509 | 0.737 |
| DE | 0.522 | 0.115 | 0.260 | 0.445 | 0.277 | 0.201 | 0.160 | 0.211 | 0.360 | 0.165 | 0.192 |
| PO | 0.920 | 0.628 | 0.758 | 0.659 | 0.753 | 0.696 | 0.871 | 0.810 | 0.773 | 0.689 | 0.646 |
| AC | 1.000 | 0.433 | 0.556 | 0.481 | 0.749 | 0.687 | 0.656 | 0.987 | 0.581 | 0.997 | 0.967 |
| VI | 0.982 | 0.212 | 0.011 | 0.158 | 0.275 | 0.523 | 0.087 | 0.101 | 0.140 | 0.218 | 0.513 |
| AM | -0.066 | -0.065 | -0.063 | -0.063 | -0.063 | -0.062 | -0.062 | -0.061 | -0.063 | -0.064 | -0.064 |

