# OpenReview forum: "Cross-Table Pretraining towards a Universal Function Space for Tabular Data"
_ICLR.cc/2026/Conference — Submitted to ICLR 2026_

### Official Review · Reviewer_G8qU · 2025-10-26

**Soundness:** 2
**Presentation:** 2
**Contribution:** 1
**Rating:** 2
**Confidence:** 5

**Summary:**

XTFormer is a pretrained Transformer-based architecture which aims to improve tabular predictions, particularly for small datasets where traditional supervised learning is prone to overfitting. It only requires a light fine-tuning for new unseen downstream tasks.

Its backbone mirrors a 4-layer FT-Transformer, except that the linear layers in the feed-forward blocks are replaced with CaLinear (Calibratable Linear) layers. Each CaLinear layer combines the outputs of M = 4 pretrained linear layers, averaged according to a weighting vector $c$. The vector $c$ is generated by a small neural network, the calibration module $M_{cal}$
as $c =M_{cal}(v)$, with $v$ a learnable vector.

XTformer follows 3 training phases:
- Cross-table pertaining
- “task calibration”: fine-tuning where the learnable vector $v$, the embedding layer as well as the output layer are optimised.
- “Refinement”: 5 epochs fine-tuning all weights.

XTFormer is compared to SAINT, AutoInt and FT-Transformer (for deep learning baselines), XGBoost and CatBoost (for Gradient Boosted trees), and to TabPFNv1.

**Strengths:**

The paper is easy to follow and adresses a timely and important problem.

**Weaknesses:**

**Major weaknesss: the paper does not compare to the state-of-the-art.**

* The deep learning baselines considered in the paper are: AutoInt, SAINT and FT-Transformer. However, several recent benchmarks [Tabarena, Talent], as well as many papers, have confirmed that these are not state-of-the-art anymore. Models such as ModernNCA [MoNCA] or RealMLP [RealMLP] obtain significantly better performances.

* Concerning pretrained baselines, the paper considers TabPFN (v1) while TabPFNv2 [TabPFNv2] has been released in early 2025, as well as other strong pretrained models such as TabICL [TabICL]. TabPFNv1 is currently significantly outperformed by both TabPFNv2 and TabICL. Moreover in the introduction l.42, the paper states that TabPFN is restricted to classification tasks, which is not true anymore since early 2025.

Overall, the paper does not cite any work published in 2024 or 2025, even though the field is evolving rapidly. This indicates that the bibliography is outdated and overlooks recent advances.

Otherwise, although the paper is clearly written, some important information regarding the experimental design are missing (see questions).

[Tabarena] Erickson, Nick, Lennart Purucker, Andrej Tschalzev, David Holzmüller, Prateek Mutalik Desai, David Salinas, and Frank Hutter. "Tabarena: A living benchmark for machine learning on tabular data." arXiv preprint arXiv:2506.16791 (2025).

[Talent] Ye, Han-Jia, Si-Yang Liu, Hao-Run Cai, Qi-Le Zhou, and De-Chuan Zhan. "A closer look at deep learning methods on tabular datasets." arXiv preprint arXiv:2407.00956 (2024).

[MoNCA] Ye, Han-Jia, Huai-Hong Yin, and De-Chuan Zhan. "Modern neighborhood components analysis: A deep tabular baseline two decades later." arXiv e-prints (2024): arXiv-2407.

[RealMLP] Holzmüller, David, Léo Grinsztajn, and Ingo Steinwart. "Better by default: Strong pre-tuned mlps and boosted trees on tabular data." Advances in Neural Information Processing Systems 37 (2024): 26577-26658.

[TabPFNv2] Hollmann, Noah, Samuel Müller, Lennart Purucker, Arjun Krishnakumar, Max Körfer, Shi Bin Hoo, Robin Tibor Schirrmeister, and Frank Hutter. "Accurate predictions on small data with a tabular foundation model." Nature 637, no. 8045 (2025): 319-326.

[TabICL] Qu, Jingang, et al. "Tabicl: A tabular foundation model for in-context learning on large data." arXiv preprint arXiv:2502.05564 (2025).

**Questions:**

- What are the datasets used for pretraining (and how many)?
- What is the definition of the different settings in the experiments (T-full, T-200, etc)? This detail is important but I did not see where it is defined.

---

### Official Review · Reviewer_Zr44 · 2025-10-31

**Soundness:** 3
**Presentation:** 3
**Contribution:** 2
**Rating:** 2
**Confidence:** 4

**Summary:**

This paper introduces XTFORMER, a cross-table pretraining framework for tabular prediction that builds a high-dimensional meta-function space to enable lightweight adaptation to new datasets. The central technical component is a novel layer called CALINEAR, which learns a small number of *basis linear layers* during large-scale pretraining. For each downstream dataset, a learned low-dimensional context vector `v` is passed through a calibration MLP (`Mcal`) to generate coefficients that linearly combine those bases, yielding dataset- and feature-specific transformations.
After pretraining across many heterogeneous tables, downstream adaptation proceeds in two stages: (1) task calibration, where only lightweight parameters (`v`, dataset-specific embeddings, output heads, and normalization layers) are tuned; and (2) refinement, a short full fine-tuning stage. Experiments on massive prediction tasks show the effectiveness of the proposed method.

**Strengths:**

- The CALINEAR mechanism (decomposing linear transformations into a learnable set of bases and calibrating them via a small context vector) is conceptually clear and technically appealing. It offers a parameter-efficient route to defining a flexible function family across datasets.

- The three-stage procedure is logically motivated and easy to follow. It clearly separates the generalization stage (pretraining) from lightweight dataset specialization (calibration) and modest full adaptation (refinement).

- The large-scale evaluation on 190 tasks from 38 datasets is commendable, representing one of the more extensive studies in tabular pretraining. The inclusion of multiple baselines and ablations suggests careful experimental effort. The authors provide ranking statistics, win/tie/loss summaries, and cross-dataset performance trends. The results appear consistent across both full and low-data regimes.

**Weaknesses:**

- Unsubstantiated theoretical claims about the “meta-function space.”
   The paper repeatedly asserts that the learned function space “encompasses all potential feature–target mappings,” yet provides neither formal justification nor empirical evidence to support this sweeping claim. There is no analysis of representational capacity, approximation error, or coverage of target functions. The term *meta-function space* remains largely metaphorical.
   → The paper would be stronger with either (a) formal grounding (e.g., approximation lemma or expressivity analysis of stacked CALINEAR + ReLU layers), or (b) tempered language reflecting this as an *empirical observation*, not a theoretical guarantee.

- Unclear design rationale for freezing `Mcal` during calibration.
   The authors freeze `Mcal` and only optimize the small context vector `v` for parameter efficiency. However, this decision seems arbitrary and possibly suboptimal. If `Mcal` poorly generalizes to some downstream feature distributions, freezing it could constrain adaptability. A comparison between freezing, partial fine-tuning, or adapter-style updates would clarify this tradeoff.

- A single shared context vector `v` (with per-feature components) is used across all CALINEAR layers. This uniformity may limit representational flexibility across network depths. The paper neither explains this design nor tests alternatives such as per-layer or hierarchical contexts, which could potentially yield richer adaptation.

- Lack of representational justification for CALINEAR stacking.
   A linear combination of linear layers is still linear; the nonlinearity arises only from ReLU or attention. The paper should provide theoretical or empirical evidence that stacking CALINEAR modules meaningfully expands representational diversity beyond ordinary feedforward networks, otherwise the “meta-function” framing feels overstated.

-  Missing ablations and sensitivity studies.
   Several important choices lack empirical exploration:

   * Initialization and dimensionality sensitivity of `v`.
   * Alternative designs of `Mcal` (softmax vs unconstrained coefficients, sparsity regularization).
   * Handling of categorical features or high-cardinality attributes.
     These omissions make it difficult to assess the robustness and generality of the approach.

- No discussion of interpretability or analysis of learned representations.
   Unlike GBDTs, which provide clear feature importance, XTFORMER’s internal mechanisms are opaque. The authors should at least discuss possible ways to interpret calibrated coefficients or attention patterns.


Suggestion:
   Phrases like “encompasses all possible mappings” and “revolutionizing tabular learning” overstate the contribution relative to the evidence. The work is promising, but the language needs restraint.

**Questions:**

- How is dataset overlap avoided between pretraining and downstream tasks? Were names, hashes, or feature overlaps checked?
- What are the total model parameters (shared transformer + CALINEAR bases + `Mcal`) and per-task calibration parameters (`v`, norms, output head)? What is the total checkpoint size?
- Why is `Mcal` frozen during calibration? Have you compared with partial fine-tuning or adapters? If not, please test this variant.
- Why use a single shared `v` instead of per-layer or hierarchical contexts? Were these alternatives explored?
- How are categorical and high-cardinality features represented? Are embeddings learned or target-encoded? How does XTFORMER compare to GBDTs in such cases?
- Why is `Mcal` output passed through a softmax (convex combination)? Would unconstrained or sparse combinations yield better flexibility? Please discuss or show sensitivity.
- Which tasks or domains show the largest performance drops versus GBDTs or FT-Transformer? Understanding failure cases would increase confidence in robustness.


I would consider raising my score if the authors can adequately address these questions.

---

### Official Review · Reviewer_dN4h · 2025-10-31

**Soundness:** 3
**Presentation:** 3
**Contribution:** 3
**Rating:** 6
**Confidence:** 4

**Summary:**

The paper proposes XFormer, a cross-table pretrained Transformer for tabular prediction that seeks to learn a transferable meta-function space rather than a single universal mapping. The core idea is to pretrain a model across diverse upstream tables to embed a variety of potential feature–target mappings into a shared space; for a new downstream task, a coordinate positioning (calibration) step selects a suitable region of this space, followed by a short refinement. On 190 downstream tasks (classification and regression), the method reportedly outperforms strong baselines: wins over XGBoost and CatBoost on 137 tasks (72%), and over deep tabular models FT-Transformer and XTab on 144 (76%) and 162 (85%) tasks, respectively. The contribution emphasizes data efficiency and broad transfer in low-label regimes, positioning cross-table pretraining as a practical path toward reusable tabular inductive biases.

**Strengths:**

**Originality**:
Reframes tabular transfer as function-space selection rather than feature/token alignment or single-table pretext training.

**Empirical quality**:
Large-scale evaluation across 190 tasks with win–tie–loss and rank-based metrics; ablations support design choices.


**Data efficiency**
Calibration tunes few parameters for new tables, aiding small-data performance and fast adaptation.

**Clarity of pipeline**: Pretraining $\rightarrow$ calibration (frozen or lightly tuned body) $\rightarrow$ short refinement is easy to implement.

**Baselines breadth**: Includes both GBDTs (XGBoost, CatBoost) and strong deep tabular models (FT-Transformer, XTab), supporting external validity.

**Weaknesses:**

**Theory gap vs. rhetoric**: The “universal/meta-function space” framing lacks formal expressivity or approximation guarantees.

**Dataset/selection bias**: Evidence is centered on public tabular suites; underexplored settings include extreme categorical cardinality, severe imbalance, temporal dependence, and distribution shift.

**Protocol clarity**: Non-overlap between pretraining and downstream sets should be documented in detail, with lists/splits for reproducibility and leakage checks.

**Budget fairness**: A matched compute/tuning budget comparison would clarify advantage sources versus tuned GBDTs and FT-Transformer.

**Robustness**: Limited analysis under covariate/label shift, OOD tables, or perturbations; failure modes are not deeply dissected.

**Questions:**

**Expressivity**: Can you provide conditions under which the pretrained space can approximate target mapping classes? Any bounds or constructive arguments?

**Calibration granularity**: How is calibration parameterized relative to feature types and counts, especially when feature dimensionality differs widely across tables?

**Leakage safeguards**: Please provide the exact dataset list, partition policy, and verification that related tables do not cross partitions.

**Compute & carbon**: What are pretraining GPU hours and carbon estimates? How do depth/width choices affect adaptation savings?

**Encoding ablations**: Sensitivity to preprocessing (quantile transforms, categorical encodings, missing-value handling) and target scaling?

**Shift robustness**: Results under covariate/label shift or OOD tables to test the “universal space” hypothesis?

**Budget matching**: Outcomes under matched tuning/compute budgets vs. GBDTs and FT-Transformer?

**Failure modes**: On the $\sim$15–30% tasks where the method underperforms, what patterns emerge (tiny $N$, extreme cardinality, strong non-linear interactions)?

---

### Official Review · Reviewer_aaj5 · 2025-11-01

**Soundness:** 2
**Presentation:** 2
**Contribution:** 2
**Rating:** 2
**Confidence:** 5

**Summary:**

This paper proposes a pretraining scheme for tabular data that constructs a meta-function space that can be shared across datasets. The work trains the basis functions and the combination of the basis functions through the neural network architecture (variant of the transformers). The experiments show solid performances compared to several baselines including gbdts and other nn architectures.

**Strengths:**

The paper is generally easy to follow. The simple, yet strong, ideas with basis functions and the ability of the XTFORMER bringing smoothness are interesting points.

**Weaknesses:**

My biggest concern with the paper is that it does not seem to provide major changes from previous submissions to machine learning conference(s). The literature review seems to remain the same, missing many of the recent advances in tabular data. Some wording, such as "revolutionizing", should be avoided and detailed explanations on how the data are processed (for instance, tokenization or categorical variables) should be included for clarity. Moreover, there should be more baselines such as linear models and TabPFNv2. In addition, the paper does not address limitations. Some limitations or further work might include computation expenses, extensions to larger body of the pretrain data, different preprocessings of numerical and categorical variables, etc.

**Questions:**

- The report of performance in the abstract is not written in the main body.
- Is there a tokenization step (as in FT transformer)? If so, does the model use the CLS token? If not, how is the self-attention calculated?
- Would the contrastive learning losses help with pretraining?
- How does the model perform in terms of direct performance comparison (not the win-rate)?
- How are the categorical features handled?
- Does the number of transformer block affect the performance?
- Are the coordinate positions same for all layers? (In figure 3, although the colors are different, the notation is fixed)
- What is the reference of $M^{2L}$?
- If it requires data-specific initial embedding layers, how would it cope with a larger body of the pretrain data?

---

### Official Review · Reviewer_rMrx · 2025-11-01

**Soundness:** 3
**Presentation:** 3
**Contribution:** 3
**Rating:** 4
**Confidence:** 3

**Summary:**

The paper introduces a new model, XTformer, which is cross-dataset pretrained on variety of datasets, and then can be task calibrated and refined on a particular dataset. The main novelty in architecture is CaLinear module that learns a linear function space and then can be adapted for a downstream task.

**Strengths:**

* Paper has a great novelty
* Good motivation and description of the method
* Insightful ablations regarding number of calibration and refinement epochs
* Overall, paper is well-written

**Weaknesses:**

See questions.

**Questions:**

1. Is it correct that the authors pretrain XTFormer and calibrate/refine using the same set (Table 8)?
2.  As far as I understand, On Figure 2 there are N features in x? How the model handles various input sizes (between datasets)? Also, on this figure there are N coefficients `c` and vector `v_N`. I think, It should be `M`? Overall, I think explanation regarding dimensions are needed.
3.  I think description of the`T-{n}`setting is missed. How XTFormer is trained in this setting? Do authors pretrain it using `n` samples from each dataset?
4. My main complaint is a weak set of baselines. GBDTs are great but the rest of the models are weak. I think it will be useful to compare XTFormer with modern DL models (e.g. from [1] or [2]). I will expect that XTFormer is weaker in T-full regime.
5. I think paper needs more comparison with TabPFN (TabPFNv2). Since the proposed model is better in few-shot regime than baselines, it is a must to compare with TabPFNv2 which is designed to work on a small datasets.

Overall, currently a paper lacks comparison with state-of-the-art DL models, like TabM or RealMLP, and with TabPFNv2 in a few-shot setting. My current score is 4 since results are not convincing, though I think the paper is useful for the field and I am ready to increase my score up to 6.

[1]: Better by Default: Strong Pre-Tuned MLPs and Boosted Trees on Tabular Data. David Holzmüller, Léo Grinsztajn, Ingo Steinwart. 2024.
[2]: TabM: Advancing tabular deep learning with parameter-efficient ensembling. Yury Gorishniy, Akim Kotelnikov, Artem Babenko . 2025.

---

### Meta-Review · Area_Chair_hc2G · 2025-12-26

**Summary:**

Reviewer consensus highlights significant concerns that temper enthusiasm for the paper's clear motivation and large-scale evaluation. The decision is primarily informed by the use of outdated baselines, with multiple reviewers noting the absence of crucial comparisons to modern SOTA methods like RealMLP, ModernNCA/TabM, TabPFNv2, and TabICL. This makes it difficult to assess the true advancement. A key factual error regarding TabPFN's capabilities—claiming it only supports classification—further undermines confidence. Additionally, the experimental protocol lacks clarity, particularly regarding the T–{n} settings, dataset splitting procedures, and safeguards against data leakage between pretraining and downstream tasks. The paper's theoretical claims about a "meta-function space" are considered unsubstantiated without formal analysis, and there is insufficient exploration of robustness, failure modes, or the handling of categorical features.

**Reviewer Concerns:**

Since no author rebuttal was submitted, none of the core concerns raised by the reviewers have been addressed. While the clarity of the method and the scale of the experimental setup are appreciated, the fundamental gaps remain outstanding. The missing comparisons to state-of-the-art models, the necessary correction of the factual statement about TabPFN, and the need for detailed clarification of the experimental protocol are all unresolved. Likewise, the disconnect between the ambitious theoretical claims and the provided evidence persists, as does the limited analysis of the model's robustness and potential failure modes.

**Reviewer Scores:**

Since the authors did not submit a rebuttal, there was no discussion phase for this submission. As a result, I do not expect any of the reviewers’ scores to change, and I consider their original scores to remain valid.

---

### Decision · Program_Chairs · 2026-01-26

Reject